# GoRA: Gradient-driven Adaptive Low Rank Adaptation

**Haonan He**[1,2,3,*], **Peng Ye**[3,4,5,*], **Yuchen Ren**[3,6], **Yuan Yuan**[2],
**Luyang Zhou**[7], **Shucun Ju**[7], **Lei Chen**[2,†]

[1]University of Science and Technology of China
[2]Institute of Intelligent Machines, HFIPS, Chinese Academy of Sciences
[3]Shanghai Artificial Intelligence Laboratory    [4]Fudan University
[5]The Chinese University of Hong Kong    [6]University of Sydney
[7]Anhui Disaster Warning & Agrometeorological Information Center
`hehn@mail.ustc.edu.cn`

## Abstract

Low-Rank Adaptation (LoRA) is a crucial method for efficiently fine-tuning large language models (LLMs), with its effectiveness influenced by two key factors: rank selection and weight initialization. While numerous LoRA variants have been proposed to improve performance by addressing one of these aspects, they often compromise usability or computational efficiency. In this paper, we analyze and identify the core limitations of existing approaches and propose a novel framework—**GoRA** (**G**radient-driven Adaptive **Lo**w **R**ank **A**daptation)—that simultaneously adapts both the rank and initialization strategy within a unified framework. GoRA leverages gradient information during training to dynamically assign optimal ranks and initialize low-rank adapter weights in an adaptive manner. To our knowledge, GoRA is the first method that not only addresses the limitations of prior approaches—which often focus on either rank selection or initialization in isolation—but also unifies both aspects within a single framework, enabling more effective and efficient adaptation. Extensive experiments across various architectures and modalities show that GoRA consistently outperforms existing LoRA-based methods while preserving the efficiency of vanilla LoRA. For example, when fine-tuning Llama3.1-8B-Base for mathematical reasoning, GoRA achieves a 5.13-point improvement over standard LoRA and even outperforms full fine-tuning by 2.05 points under high-rank settings. Code is available at: https://github.com/hhnqqq/MyTransformers.

## 1 Introduction

Open-source pre-trained large language models (LLMs) such as the Llama series [1, 2] have demonstrated exceptional capabilities. Through supervised fine-tuning, these models can be adapted to various downstream tasks such as code generation [3] and mathematical problem solving [4]. However, when the model has a parameter size $\phi$ and uses FP16/BF16 mixed-precision training strategy [5, 6] with the Adam optimizer [7], the parameters and gradients require $4\phi$ bytes of memory, while the optimizer states require $12\phi$ bytes. Thus, the minimum memory usage, excluding activations, reaches $16\phi$ bytes. Such high memory demands limit the training of large language models under constrained resources. To reduce memory usage, Low-Rank Adaptation (LoRA) [8] decomposes the weight matrix $\mathbf{W} \in \mathbb{R}^{m \times n}$ into $\mathbf{W} = \mathbf{W}_0 + \Delta\mathbf{W} = \mathbf{W}_0 + s\mathbf{AB}$, where $s$ is a scaling factor, and $\mathbf{A} \in \mathbb{R}^{m \times r}, \mathbf{B} \in \mathbb{R}^{r \times n}, r \ll \min(m, n)$, as shown in Figure 1(a). LoRA only updates the low-rank weights $\mathbf{A}$ and $\mathbf{B}$, keeping the pre-trained weight $\mathbf{W}_0$ unchanged, thereby significantly reducing the memory footprint of optimizer states. Although LoRA performs well on simple tasks, when applied to pre-trained large language models, its performance on more challenging tasks, such as mathematical reasoning and code generation, still lags behind full fine-tuning [9, 10].

---

[*]Equal contribution.

[†]Corresponding author: chenlei@iim.ac.cn.

39th Conference on Neural Information Processing Systems (NeurIPS 2025).

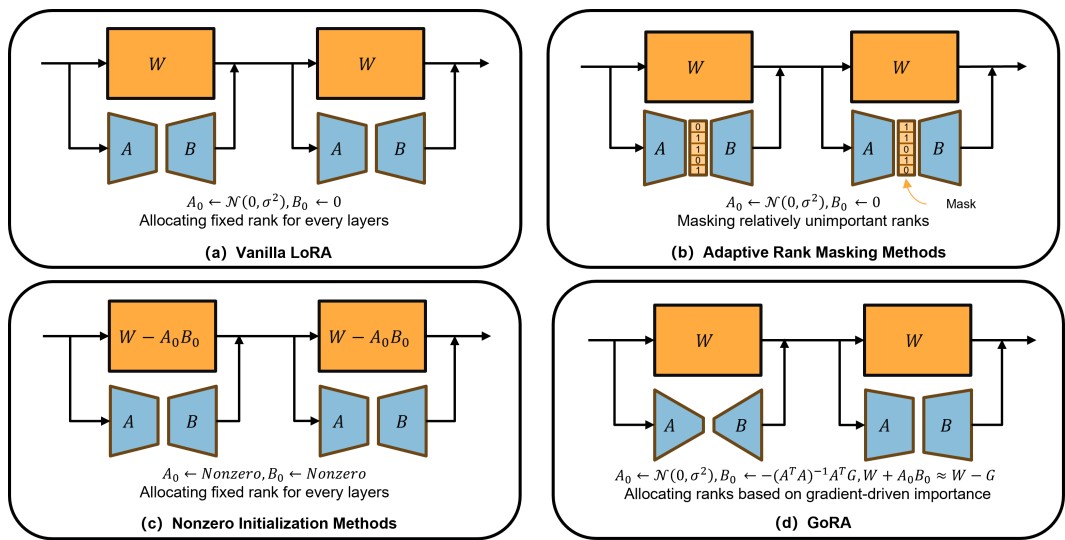

Figure 1: Illustration of (a) LoRA; (b) LoRA variants utilizing adaptive rank masking strategies; (c) LoRA variants employing nonzero initialization strategies; and (d) GoRA, which introduces adaptively leveraging the weight $\mathbf{W}$'s gradient to allocate the rank of the low-rank adapter and initialize $\mathbf{B}$. $\mathbf{A}_0$ and $\mathbf{B}_0$ denote the initialized value of $\mathbf{A}$ and $\mathbf{B}$.

One of the critical factors in LoRA is its rank. Kalajdzievski [11] demonstrates that increasing the rank of LoRA can significantly improve performance when paired with an appropriate scaling factor. However, a direct increase in rank leads to a substantial rise in memory requirement overhead, thus imposing constraints on rank selection. To address this, several studies [12, 13] propose to ensemble multiple low-rank subspaces, allowing for rank increases without proportionally increasing the number of trainable parameters. Nevertheless, these approaches often come at the expense of usability due to their intrusion into architecture or training processes. Another promising line of research explores adaptively assigning ranks to pre-trained weights based on importance. For example, AdaLoRA [14] adaptively adjusts ranks by quantifying the importance of each rank during training and masking less significant ones, as illustrated in Figure 1(b). However, this masking mechanism necessitates a larger parameter space (*e.g.*, 1.5 times), increasing the number of trainable parameters and limiting the upper bound of rank. Consequently, as demonstrated in Section 5.1, adaptively allocating ranks without significantly increasing the training cost remains an open challenge.

Another vital factor in LoRA is its initialization strategy. In vanilla LoRA, $\mathbf{A}_0$ is initialized with a normal distribution (In the PEFT library, $\mathbf{A}_0$ is initialized with a Kaiming distribution [15]) and $\mathbf{B}_0$ is initialized with zeros. This initialization method ensures that the weights $\mathbf{W}_0 + \mathbf{A}_0\mathbf{B}_0$ remain unchanged at the beginning of training. Besides, zero initialization is not the only option: When $\mathbf{A}_0\mathbf{B}_0$ is nonzero, manipulating the pre-trained weight by subtracting $\mathbf{A}_0\mathbf{B}_0$ from $\mathbf{W}_0$ also ensures stability. Existing nonzero initialization methods can be categorized into experience-driven and data-driven methods. In experience-driven methods, PiSSA [16] and MiLoRA [17] employ decomposition techniques such as Singular Value Decomposition (SVD) to capture specific features of pre-trained weights. However, these methods are inherently task-agnostic, which limits their generalizability across diverse tasks. In contrast, data-driven methods incorporate task information. For example, LoRA-GA [18] uses the singular features of gradients to initialize LoRA weights, minimizing the difference between LoRA and full fine-tuning. However, as illustrated in Figure 1(c) and Section 2.2, existing nonzero methods require manipulating the pre-trained weights, resulting in a training-inference gap. Thus, designing a nonzero initialization method without manipulating pre-trained weights remains an open problem.

Given the challenges of adaptive rank allocation and nonzero initialization, we turn to gradients of pre-trained weights, which are crucial for assessing the importance of pre-trained weights and deeply related to LoRA adapters' optimization processes [19, 20]. As shown in Figure 1(d), we propose GoRA. Specifically, before training, we compute gradients of pre-trained weights on a subset of training samples, using these gradients to assess the importance of each pre-trained weight. Given a reference rank, we calculate a trainable parameter budget. Based on the normalized importance and the trainable parameter budget, we allocate a new trainable parameter count and corresponding rank for

each low-rank adapter, achieving adaptive rank allocation without significantly increasing the trainable parameter count compared to LoRA and allowing for higher rank allocation upper bounds, as shown in Table 5. In GoRA's initialization, we maintain LoRA's initialization for $\mathbf{A}$, while $\mathbf{B}$ is initialized using $-(\mathbf{A}_0^\top \mathbf{A}_0)^{-1} \mathbf{A}_0^\top \mathbf{G}$, where $\mathbf{G}$ is the gradient of the weight $W$. This initialization ensures that the computation result of the low-rank adapter $-\mathbf{A}_0(\mathbf{A}_0^\top \mathbf{A}_0)^{-1} \mathbf{A}_0^\top \mathbf{G} \approx -\mathbf{G}$ compresses the gradient optimally, setting a solid foundation for further optimization without manipulating the pre-trained weight. Our key contributions are summarized as follows:

1. We conduct an in-depth investigation into LoRA's rank allocation and initialization strategy, uncovering the limitations of existing works. We propose GoRA, which achieves adaptive rank allocation and initialization without compromising usability and efficiency.

2. We use the gradients of weights to assess their importance and allocate ranks. We then initialize the low-rank weights using the pseudo-inverse of the compressed gradients, which enhances performance while ensuring training stability.

3. We conduct extensive experiments, demonstrating that GoRA consistently outperforms low-rank baselines and even rivals full fine-tuning in certain settings. For example, on the Llama3.1-8B-Base model fine-tuned for mathematical reasoning, GoRA achieves a 5.13-point improvement over LoRA and even surpasses full fine-tuning high-rank configurations.

## 2 Related Works

### 2.1 Rank of LoRA

The choice of rank is crucial for LoRA, with higher ranks consistently yielding better outcomes [11]. However, increasing the rank raises the number of trainable parameters and corresponding memory usage overhead, making it challenging to train with sufficient ranks on limited hardware resources. Previous works [21, 12] attempt to continuously merge and reinitialize low-rank weights during training to stack the overall rank. However, these methods often require resetting the states of the optimizer and learning rate scheduler during reinitialization to ensure that updates take place in distinct low-rank subspaces and ensure training stability, significantly increasing overall training complexity and making the training process unstable. MeLoRA [13] proposes aggregating multiple mini low-rank adapters diagonally to increase the overall rank. Nevertheless, this approach requires modifying the structure of LoRA, limiting its usability.

At the same time, the significances of weights during training demonstrate heterogeneity, and an intuitive proposition is to assign larger ranks to relatively more important weights. Previous works [14, 22] attempted to dynamically mask less important ranks during training to achieve adaptive rank adjusting. However, these methods require allocating larger matrices for low-rank adapters to reserve space for masked ranks, leading to an increase in the number of trainable parameters, which compromises their operational efficacy and establishes limitations on the upper threshold of rank. IncreLoRA [23] introduces an approach that begins with a single rank for each low-rank adapter and incrementally increases the rank during training. This method effectively addresses the challenge of large initial matrices. Nevertheless, this approach demonstrates suboptimal compatibility with distributed training architectures, notably FSDP[24] and ZeRO[25], which constitute essential infrastructural components for the effective training of large-scale models.

### 2.2 Initialization of LoRA

Parameter initialization represents a fundamental paradigm in deep learning methodologies. Well-established initialization protocols, such as the strategy proposed by Xavier Glorot and Yoshua Bengio [26], facilitate the convergent training trajectories of deep neural networks. Similarly, appropriate initialization strategies constitute a critical determinant for LoRA. Beyond zero initialization used by vanilla LoRA, some studies have explored different initialization strategies: PiSSA [16] performs SVD on pre-trained weights and uses the most important singular features to initialize low-rank weights; MiLoRA [17], in contrast to PiSSA, uses the least important singular features to initialize low-rank weights; similarly, OLoRA [27] uses QR decomposition of pre-trained weights to initialize low-rank weights; EVA [28] uses singular features of activations to initialize low-rank weights; and LoRA-GA [18] uses singular features of gradients to initialize low-rank weights. These methods can improve LoRA's performance to some extent.

Nevertheless, owing to the inherent non-zero initialization characteristics of these methodologies, they require subtracting the LoRA initialization results from the pre-trained weights to ensure correct forward and backward propagation during the initial phases of the training regimen, consequently creating a gap between training and inference. Recomputing the initialization result of these methods during inference is not feasible in cases involving randomness [16] or requiring original training data [18, 28]. And the initialization process requires significant time for methods such as MiLoRA [17]. The most straightforward solution is to save not only the low-rank weights but also the manipulated pre-trained weights, but this sacrifices one of LoRA's significant advantages, namely, minimal checkpoint storage [29]. Another approach is to save the initialized LoRA weights and use block matrix multiplication to eliminate the gap, but this reduces usability.

## 3 Method

In this section, we will reinterpret LoRA adapters from the perspective of gradient compressors and introduce GoRA's gradient-driven adaptive rank allocation and initialization strategy.

### 3.1 View LoRA adapters as Gradient Compressors

The core idea of LoRA is to fine-tune a model by leveraging the intrinsic low-rank property of the update of a weight matrix $\mathbf{W} \in \mathbb{R}^{m \times n}$ during training. Specifically, a pair of low-rank matrices $\mathbf{A} \in \mathbb{R}^{m \times r}$ and $\mathbf{B} \in \mathbb{R}^{r \times n}$ are initialized alongside the pre-trained weight $\mathbf{W}_0$. During training, $\mathbf{W}_0$ remains frozen, while the model is updated by training the low-rank matrices $\mathbf{A}$ and $\mathbf{B}$, thereby reducing memory usage during training. For any training step $t$, the updated weight $\mathbf{W}_t$ is given by (1), where $\alpha$ is a tunable hyperparameter that ensures the scale of the LoRA computation depends only on $\alpha$ and is independent of the rank $r$:

$$\mathbf{W}_t = \mathbf{W}_0 + \Delta \mathbf{W}_t = \mathbf{W}_0 + \frac{\alpha}{r} \mathbf{A}_t \mathbf{B}_t. \tag{1}$$

Specifically, given the training loss $\mathcal{L}$, the gradient of the pre-trained weight $\mathbf{W}_0$ can be computed as $\frac{\partial \mathcal{L}}{\partial \mathbf{W}_0}$. Using the chain rule, the gradients of $\mathbf{A}$ and $\mathbf{B}$ are $\frac{\partial \mathcal{L}}{\partial \mathbf{W}_0} \mathbf{B}_t^\top$ and $\mathbf{A}_t^\top \frac{\partial \mathcal{L}}{\partial \mathbf{W}_0}$, respectively. (Note: in vanilla LoRA, $\mathbf{B_0} = 0$.) Given a learning rate $\eta$, the updates to the weight are as shown in (2)-(3):

$$\Delta \mathbf{B}_t = -\eta \frac{\alpha}{r} \sum_{t=1}^{T} \mathbf{A}_{t-1}^\top \frac{\partial \mathcal{L}_t}{\partial \mathbf{W}_0}, \quad \Delta \mathbf{A}_t = -\eta \frac{\alpha}{r} \sum_{t=1}^{T} \frac{\partial \mathcal{L}_t}{\partial \mathbf{W}_0} \mathbf{B}_{t-1}^\top, \tag{2}$$

$$\Delta \mathbf{W}_t = \frac{\alpha}{r} \mathbf{A}_t \mathbf{B}_t - \frac{\alpha}{r} \mathbf{A}_0 \mathbf{B}_0 = \frac{\alpha}{r} (\Delta \mathbf{A}_t \Delta \mathbf{B}_t + \mathbf{A}_0 \Delta \mathbf{B}_t). \tag{3}$$

Experimental results from LoRA-FA [30] have shown that freezing the randomly initialized matrix $\mathbf{A}$ and only training the matrix $\mathbf{B}$ can achieve performance close to that of LoRA. When matrix $\mathbf{A}$ is frozen ($\Delta \mathbf{A}_t = 0$), the weight update is given by (4). One can observe that matrix $\mathbf{B}$ accumulates the gradients compressed by $\mathbf{A}_0^\top$ during training, and when multiplied by $\mathbf{A}_0$, the compressed gradients are up-projected. Thus, the training process of LoRA-FA can be viewed as a process of gradient accumulation and compression, with the compression matrix being the randomly initialized $\mathbf{A}$.

$$\Delta \mathbf{W}_t = \frac{\alpha}{r} \mathbf{A}_0 \Delta \mathbf{B}_t = -\eta \frac{\alpha}{r} \sum_{t=0}^{T} \mathbf{A}_0 \mathbf{A}_0^\top \frac{\partial \mathcal{L}_t}{\partial \mathbf{W}_0}. \tag{4}$$

The update form of LoRA-FA provides significant inspiration. We hypothesize that vanilla LoRA has similar properties, i.e., LoRA adapters act as gradient compressors. Based on this hypothesis, we can allocate larger ranks to weights whose gradients and weights themselves contain more low-rank information and initialize LoRA parameters using compressed gradients.

### 3.2 GoRA's Adaptive Rank Allocation Strategy

Based on the hypothesis that LoRA adapters function similarly to gradient compressors, our adaptive rank allocation strategy aims to: (1) allocate rank based on weight importance derived from pre-computed $N$-batch accumulated gradients $\mathbf{G} = \frac{1}{N} \sum_{i=1}^{n} \frac{\partial \mathcal{L}_i}{\partial \mathbf{W_0}}$ before the formal training process; (2) complete rank allocation before training to avoid dynamic shape changes; (3) maintain a similar number of trainable parameters as LoRA (within 10%); and (4) preserve structural compatibility with LoRA for easy integration.

To evaluate the importance of weights, we first consider the nuclear norm of the pre-computed gradient, which aggregates all singular values of a matrix and is often used to measure low-rank properties [18]. However, as shown in Table 6, this metric does not effectively capture the importance of weights in practice. Instead, we adopt a sensitivity-based importance metric commonly used in model pruning [31]. Specifically, we define the importance of a weight matrix $\mathbf{W}$ as:

$$\mathrm{I}(\mathbf{W}) = \mathrm{avg}\left(|\mathbf{W} \odot \mathbf{G}|\right), \tag{5}$$

where the operator $\odot$ denotes the Hadamard product and $\mathrm{avg}(\cdot)$ computes the average value to yield a scalar importance score. After computing the importance scores for all target pre-trained weight matrices, we form an importance set $\{\mathrm{I}(\mathbf{W}^i)\}_{i=1}^N$. To facilitate adaptive rank allocation, we normalize the importance set to compute an advantage $a_i$ for the $i$-th pre-trained weight $\mathbf{W}_0^i$:

$$a^i = \frac{\mathrm{I}(\mathbf{W}_0^i)}{\Sigma_{i=1}^N \mathrm{I}(\mathbf{W}_0^i)}. \tag{6}$$

With the normalized advantages computed, we next determine the total trainable parameter budget $b$ for the model. Given a reference rank $r^{\mathrm{ref}}$, the budget for a single weight matrix $\mathbf{W}^i \in \mathbb{R}^{m \times n}$ is estimated as:

$$b^i = (\sqrt{m+n}) \times r^{\mathrm{ref}}, \tag{7}$$

reflecting a smoothed parameter budget under vanilla LoRA. Summing over all budgets, the total budget becomes $b = \Sigma_{i=1}^N b^i$. Using this budget and the advantage $a^i$, we allocate the adapter rank $r^i$ and its trainable parameter count $p^i$ for each weight matrix as:

$$r^i = \left[\frac{p^i}{\sqrt{m+n}}\right] = \left[\frac{b * a^i}{\sqrt{m+n}}\right], \quad \mathrm{s.t.} r^{\mathrm{min}} \le r^i \le r^{\mathrm{max}}, \tag{8}$$

where the operator $[\cdot]$ denotes rounding to the nearest integer, and $r^{\mathrm{min}}, r^{\mathrm{max}}$ are hyper-parameters defining the allowable rank range. This formulation ensures that the total number of trainable parameters $p = \Sigma_{i=1}^N p^i$ approximately matches that of standard LoRA with rank $r^{\mathrm{ref}}$ closely.

In summary, before training begins, we use the $N$-batch accumulated gradients for all target weights. These gradients are then used to estimate the importance of each weight matrix, based on which we perform adaptive rank allocation in GoRA, achieving all four objectives outlined earlier.

### 3.3 GoRA's Adaptive Initialization Strategy

Once ranks are allocated for each layer, it is crucial to initialize the low-rank weights properly. The compression form in (4) is suboptimal when $\mathbf{A}$ is randomly initialized and fixed; to achieve better alignment with the gradient dynamics, we can initialize $\mathbf{B}$ such that the computation of the low-rank adapter at the start of training closely approximates the $N$-batch accumulated gradient $\mathbf{G}$. This optimal initialization can be derived using the Moore-Penrose inverse of $\mathbf{A}_0$ (this computation requires negligible computational time as detailed in Appendix D.1):

$$\mathbf{B}_0 = -(\mathbf{A}_0^\top \mathbf{A}_0)^{-1}\mathbf{A}_0^\top \mathbf{G}, \quad \mathbf{A_0 B_0} = -\mathbf{A}_0(\mathbf{A}_0^\top \mathbf{A}_0)^{-1}\mathbf{A}_0^\top \mathbf{G}. \tag{9}$$

As shown in (9), initializing $\mathbf{B}$ as $-(\mathbf{A}_0^\top \mathbf{A}_0)^{-1}\mathbf{A}_0^\top \mathbf{G}$ ensures that $\mathbf{A_0 B_0}$ provides the best low-rank approximation of $\mathbf{G}$ given a fixed $\mathbf{A}_0$, with proof provided in Appendix B.1.

However, due to the properties of pseudo-inverse computation, the magnitude of $\mathbf{A}_0\mathbf{B}_0$ does not exactly match that of $\mathbf{G}$. Assuming both $\mathbf{G} \in \mathbb{R}^{m \times n}$ and $\mathbf{A}_0 \in \mathbb{R}^{m \times r}$ follow distributions with mean 0 and variance 1, the expected Frobenius norm of $\mathbf{G}$, $\mathbb{E}[||\mathbf{G}||_F]$, is $\sqrt{mn}$, while that of $\mathbf{A}_0\mathbf{B}_0$, $\mathbb{E}[||\mathbf{A}_0\mathbf{B}_0||_F]$, is $\sqrt{rn}$, as detailed in Appendix B.2.

To ensure that the initial computation of a low-rank adapter approximates a single step of stochastic gradient descent with a tunable step size $\gamma$, we introduce a scaling factor $\xi$ for $\mathbf{B}_0$:

$$\frac{\alpha}{r}\mathbf{A}_0(\xi \mathbf{B}_0) \approx \xi \frac{\alpha}{r}\sqrt{\frac{r}{m}}\mathbf{G} \approx -\gamma\mathbf{G}. \tag{10}$$

Thus, to make GoRA's initialization equivalent to one step of gradient descent, $\xi$ should be set to $\frac{\gamma \cdot \sqrt{rm}}{\alpha}$. Inspired by rsLoRA [11], and to better utilize the larger ranks obtained through dynamic allocation, we modify the forward computation to $\mathbf{W}_t = \mathbf{W}_0 + \Delta\mathbf{W} = \mathbf{W}_0 + \frac{\alpha}{\sqrt{r}}\mathbf{A}_t\mathbf{B}_t$, which adjusts $\xi$ to $\frac{\gamma \cdot \sqrt{m}}{\alpha}$. Setting $\gamma$ to a relatively large value further improves performance and yields optimal results. The full algorithm is summarized in Algorithm 1 and Algorithm 2.

---

**Algorithm 1** Rank Allocation and Initialization of GoRA under Single Training Worker

---

**Input:** Model $f(\cdot)$ with $L$ layers, pre-trained parameters $\theta = \{\mathbf{W}_0^l\}_{l=1}^L$, gradient accumulation steps $N$, loss function $\mathcal{F}$, scale factor $\gamma$, Trainable parameter budget $b$

**Output:** Initialized low-rank matrices $\{\mathbf{A}_0^l\}_{l=1}^L$, $\{\mathbf{B}_0^l\}_{l=1}^L$

1: **for** $l = 1$ to $L$ **do**
2:     $\mathbf{G}_{\text{avg}}^l \leftarrow 0$                            ▷ Initialize gradients buffer in CPU memory.
3: **for** $i = 1$ to $N$ **do**                     ▷ Compute gradients without optimizer.
4:     Randomly sampled mini-batch $\{x, y\}$
5:     $\hat{y} \leftarrow f(x, \theta)$, $\mathcal{L} \leftarrow \mathcal{F}(y, \hat{y})$
6:     **for** $l = 1$ to $L$ **do**
7:        Accumulate on CPU: $\mathbf{G}_{\text{avg}}^l \leftarrow \mathbf{G}_{\text{avg}}^l + \frac{1}{N}\frac{\partial \mathcal{L}}{\partial \mathbf{W}_0^l}$
8: **for** $l = 1$ to $L$ **do**
9:     Compute importance $I(\mathbf{W}_0^l) \leftarrow \text{avg}(|\mathbf{W}_0^l * \mathbf{G}_{\text{avg}}^l|)$
10: **for** $l = 1$ to $L$ **do**
11:     Compute advantage $a^l \leftarrow \frac{I(\mathbf{W}_0^l)}{\sum_{l=1}^L I(\mathbf{W}_0^l)}$
12: **for** $l = 1$ to $L$ **do**
13:     $m, n \leftarrow \text{size}(\mathbf{W}_0^l)$
14:     $r^l \leftarrow \text{clip}(\text{round}(\frac{b \cdot a^l}{\sqrt{m+n}}), r^{\min}, r^{\max})$       ▷ Clip by $r^{min}$ and $r^{max}$ to avoid extremum.
15:     $\mathbf{A}_0^l \sim \mathcal{U}_{\text{Kaiming}}(m \times r^l, \ a = \sqrt{5})$     ▷ Initialize $\mathbf{A}_0^l \in \mathbb{R}^{m \times r^l}$ with Kaiming uniform.
16:     $\mathbf{B}_0^l \leftarrow -(\mathbf{A}_0^{l\top}\mathbf{A}_0^l)^{-1}\mathbf{A}_0^{l\top}\mathbf{G}_{\text{avg}}^l$             ▷ Initialize $\mathbf{B}_0^l \in \mathbb{R}^{r^l \times n}$.
17:     $\mathbf{B}_0^l \leftarrow \frac{\gamma\sqrt{m}}{\alpha}\mathbf{B}_0^l$
18: **Return** $\{\mathbf{A}_0^l\}_{l=1}^L$, $\{\mathbf{B}_0^l\}_{l=1}^L$

---

## 4 Experiments

We conducted comprehensive experiments comparing GoRA with baseline methods on natural language understanding (Section 4.1), generation tasks (Section 4.2), and image classification tasks (Section 4.3). For understanding tasks, we trained T5-Base [32] on five tasks of GLUE [33] (MNLI, SST-2, CoLA, QNLI, MRPC) and reported accuracy on corresponding validation sets. For generation tasks, we fine-tuned Llama-3.1-8B-Base [2] and Llama-2-7B-Base [1] on chat, mathematics, and coding datasets, evaluating test performance on MTBench [34], GSM8k [35], and HumanEval [36]. For image classification tasks, we fine-tuned CLIP-ViT-B/16 [37] on seven datasets including Stanford-Cars [38], DTD [39], EuroSAT [40], GT-SRB [41], RESISC45 [42], SUN397 [43] and SVHN [44] and reported test accuracy. All experiments were conducted using single-epoch training across three random seeds, with results reported as mean values and standard deviations. Unless specified otherwise, we set the LoRA rank or GoRA's reference rank $r^{\text{ref}}$ to 8. The hyperparameters of GoRA are detailed in Appendix C.3.

### 4.1 Experimental Results on Natural Language Understanding Tasks

Table 1: Performance of fine-tuning T5-Base on 5 sub-tasks of the GLUE benchmark. **Bold** and underline indicate the highest and second-highest scores of low-rank methods with $r = 8$ or $r^{\text{ref}} = 8$.

| Method | MNLI | SST-2 | CoLA | QNLI | MRPC | Average |
|---|---|---|---|---|---|---|
| Full | 86.33±0.00 | 94.75±0.21 | 80.70±0.24 | 93.19±0.22 | 84.56±0.73 | 87.91 |
| LoRA [8] | 85.30±0.04 | 94.04±0.11 | 69.35±0.05 | 92.96±0.09 | 68.38±0.01 | 82.08 |
| *Convergence Optimization Methods for LoRA* | | | | | | |
| rsLoRA [11] | 85.73±0.10 | 94.19±0.23 | 72.32±1.12 | 93.12±0.09 | 52.86±2.27 | 79.64 |
| DoRA [45] | 85.67±0.09 | 94.04±0.53 | 72.04±0.94 | 93.04±0.06 | 68.08±0.51 | 82.57 |
| LoRA+ [46] | 85.81±0.09 | 93.85±0.24 | 77.53±0.20 | 93.14±0.03 | 74.43±1.39 | 84.95 |
| *Initialization Optimization Methods for LoRA* | | | | | | |
| PiSSA [16] | 85.75±0.07 | 94.07±0.06 | 74.27±0.39 | 93.15±0.14 | 76.31±0.51 | 84.71 |
| LoRA-GA [18] | 85.70±0.09 | 94.11±0.18 | **80.57±0.20** | 93.18±0.06 | 85.29±0.24 | 87.77 |
| *Adaptive Methods for LoRA* | | | | | | |
| AdaLoRA [14] | 85.45±0.11 | 93.69±0.20 | 69.16±0.24 | 91.66±0.05 | 68.14±0.28 | 81.62 |
| GoRA | **85.91±0.02** | **94.68±0.43** | 79.86±0.35 | **93.27±0.08** | **86.10±0.20** | **87.96** |

**Settings**: We adopted baseline performances reported by LoRA-GA [18], maintaining their experimental parameters for fair comparison: Adam[7] optimizer ($\beta_1 = 0.9, \beta_2 = 0.999$, weight decay =

0), batch size 32, cosine decay learning rate with a warmup ratioo of 0.03. We trained all linear layers except the language head using a peak learning rate of 1e-4, a maximum sequence length of 128, and FP32 precision.

**Results**: Table 1 compares GoRA against multiple baselines across five GLUE benchmark tasks. GoRA achieved superior performance on four datasets (MNLI, SST-2, QNLI, and MRPC), demonstrating exceptional adaptability and generalization. While slightly underperforming LoRA-GA on CoLA by just 0.71 percentage points, GoRA's average score (87.96) surpassed all baselines and even exceeded full fine-tuning (87.91). This confirms GoRA's ability to maximize model potential while maintaining parameter efficiency. Notably, GoRA showed particularly strong performance on MRPC and QNLI, highlighting its effectiveness in small-sample learning and sentence-pair tasks.

### 4.2 Experimental Results on Natural Language Generation Tasks

**Settings**: We trained mathematical, coding, and dialogue capabilities using 100K MetamathQA [47], 100K Code-FeedBack [48] (code-only labels), and 52K WizardLM [49] subsets, respectively. For experiments on Llama-3.1-8B-base, training used AdamW [50] ($\beta_1 = 0.9, \beta_2 = 0.999$, weight decay $= 5e-4$) with batch size 64, cosine decay learning rate (warmup ratio=0.03, decay ratio=0.1), and BF16 mixed precision. For all methods, including GoRA, we trained attention modules' linear components with a peak learning rate of 5e-5 (5e-4 for AdaLoRA). Evaluation metrics: mathematics—regex-extracted accuracy; coding—PASS@1; dialogue—average scores (0-10) from GPT-4o [51], Gemini-1.5-Pro [52], and Llama-3.1-70B-Instruct [2] using prompts from [34]. For experiments on Llama-2-7B-Base, we adopted baseline results from LoRA-GA [18], and we maintained the same training and evaluation settings. Further details are provided in Appendix C.4.

Table 2: Performance of fine-tuning Llama-3.1-8B-Base.

| Method | MTBench | GSM8k | HumanEval |
|---|---|---|---|
| Full | 5.88±0.23 | 73.69±0.28 | 51.63±1.27 |
| LoRA [8] | 6.15±0.02 | 67.78±1.25 | 43.09±0.35 |
| rsLoRA [11] | 6.18±0.09 | 68.36±0.74 | 45.78±2.80 |
| DoRA [45] | 6.24±0.12 | 69.17±1.00 | 43.70±1.54 |
| LoRA+ [46] | **6.35±0.10** | 71.29±0.93 | 44.51±2.11 |
| OLoRA [27] | 6.13±0.04 | 68.54±0.42 | 43.29±2.44 |
| PiSSA [16] | 6.08±0.09 | 68.56±1.03 | 44.10±1.54 |
| LoRA-GA [18] | 5.99±0.06 | 71.39±0.90 | 43.29±0.61 |
| AdaLoRA [14] | 6.19±0.16 | 70.63±0.77 | 41.46±3.66 |
| GoRA | 6.34±0.04 | **72.91±0.76** | **48.98±2.14** |
| GoRA$_{r^{\text{ref}}=32}$ | 6.21±0.10 | 75.59±1.04 | 51.22±1.83 |
| GoRA$_{r^{\text{ref}}=128}$ | 5.82±0.31 | 75.74±0.40 | 52.03±1.41 |

Table 3: Performance of fine-tuning Llama-2-7B-Base.

| Method | MTBench | GSM8k | HumanEval |
|---|---|---|---|
| Full | 5.30 ± 0.11 | 59.36 ± 0.85 | 35.31 ± 2.13 |
| LoRA [8] | 5.61 ± 0.10 | 42.08 ± 0.04 | 14.76 ± 0.17 |
| rsLoRA [11] | 5.25 ± 0.03 | 45.62 ± 0.10 | 16.01 ± 0.79 |
| DoRA [45] | **5.97 ± 0.02** | 53.07 ± 0.75 | 19.75 ± 0.41 |
| LoRA+ [46] | 5.71 ± 0.08 | 52.11 ± 0.62 | 18.17 ± 0.52 |
| OLoRA [27] | 5.30 ± 0.04 | 43.29 ± 0.83 | 17.22 ± 0.12 |
| PiSSA [16] | 5.30 ± 0.02 | 44.54 ± 0.27 | 16.02 ± 0.17 |
| LoRA-GA [18] | 5.95 ± 0.16 | 53.60 ± 0.30 | 19.81 ± 1.46 |
| AdaLoRA [14] | 5.57 ± 0.05 | 50.72 ± 1.39 | 17.80 ± 0.44 |
| GoRA | 5.61 ± 0.12 | **54.04 ± 0.22** | **24.80 ± 1.04** |
| GoRA$_{r^{\text{ref}}=32}$ | 5.75 ± 0.06 | 56.18 ± 0.10 | 26.83 ± 2.84 |
| GoRA$_{r^{\text{ref}}=128}$ | 6.05 ± 0.04 | 56.58 ± 0.12 | 27.85 ± 0.58 |

**Results**: Table 2 and Table 3 show the performance of GoRA and baseline methods on fine-tuned Llama3.1-8B-Base and Llama2-7B-Base. Specifically, GoRA demonstrated exceptional performance on the more challenging HumanEval and GSM8k benchmarks, substantially surpassing all baseline methods. For Llama3.1-8B-Base, on the GSM8k dataset, GoRA scored 72.91, outperforming LoRA-GA's 71.39 by 1.52 points; on the HumanEval dataset, GoRA achieved 48.98, surpassing rsLoRA's 45.78 by 3.20 points. On MTBench, GoRA slightly underperforms in terms of overall effectiveness, scoring 6.34, just 0.01 points lower than LoRA+'s 6.35. Notably, GoRA performed well across different rank allocation settings. For example, GoRA$_{r^{\text{ref}}=128}$ achieved 75.74 and 52.03 on the GSM8k and HumanEval, respectively, surpassing full fine-tuning's 73.69 and 51.63. Even the $r^{\text{ref}} = 32$ configuration of GoRA, while slightly underperforming $r^{\text{ref}} = 128$, still outperformed full fine-tuning on GSM8k. For Llama2-7B-Base, GoRA demonstrated similar superior results compared to baseline methods. Especially, GoRA outperformed LoRA-GA by 4.99

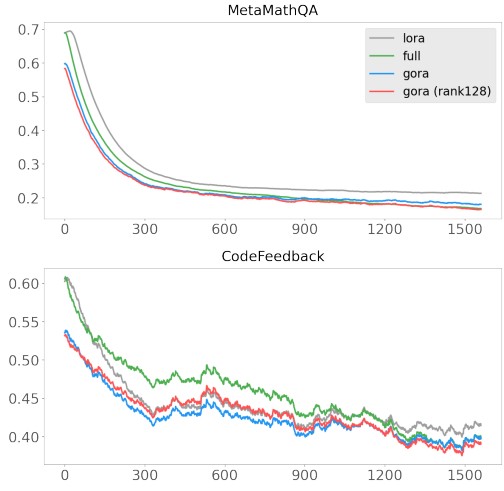

Figure 2: The training loss curves of full fine-tuning, LoRA, GoRA, and GoRA$_{r_0=128}$ on Llama-3.1-8B-Base. GoRA demonstrates lower start loss and faster convergence speed.

and 0.44 on HumanEval and GSM8k, respectively. These results validate the effectiveness of GoRA across different LLMs and settings. The training loss curves of GoRA are depicted in Figure 2.

Table 4: Performance of fine-tuning CLIP-VIP-B/16 on 7 image classification tasks.

| Method | Cars | DTD | EuroSAT | GTSRB | RESISC45 | SUN397 | SVHN | Average |
|---|---|---|---|---|---|---|---|---|
| Zero-shot | 63.75 | 44.39 | 42.22 | 35.22 | 56.46 | 62.56 | 15.53 | 45.73 |
| Full | 84.23±0.06 | 77.44±0.19 | 98.09±0.03 | 94.31±0.28 | 93.95±0.00 | 75.35±0.10 | 93.04±0.18 | 88.06 |
| LoRA [8] | 72.81±0.13 | 73.92±0.38 | 96.93±0.07 | 92.40±0.10 | 90.03±0.14 | 70.12±0.18 | 88.02±0.07 | 83.46 |
| DoRA [45] | 73.72±0.06 | 73.72±0.33 | 96.95±0.01 | 92.38±0.08 | 90.03±0.08 | 70.20±0.19 | 88.23±0.05 | 83.48 |
| LoRA+ [46] | 72.87±0.18 | 74.07±0.45 | 97.01±0.02 | 92.42±0.18 | 89.96±0.11 | 70.17±0.15 | 88.08±0.05 | 83.51 |
| LoRA-Pro [53] | **85.87±0.08** | **78.64±0.85** | 98.46±0.03 | 95.66±0.05 | 94.75±0.21 | 76.42±0.14 | 94.63±0.20 | 89.20 |
| LoRA-GA [18] | 85.18±0.41 | 77.50±0.12 | 98.05±0.27 | 95.28±0.10 | 94.43±0.19 | 75.44±0.06 | 93.68±0.35 | 88.51 |
| GoRA | 85.76±0.19 | 78.17±0.32 | **98.77±0.35** | **96.66±0.36** | **95.16±0.26** | **76.46±0.08** | **95.32±0.13** | **89.47** |

## 4.3 Experimental Results on Image Classification Tasks

**Settings**: We adopted baseline performances from LoRA-Pro [53], maintaining their experimental hyper-parameters for a fair comparison: Adam [7] optimizer ($\beta_1 = 0.9, \beta_2 = 0.999$, weight decay = 0), batch size 64, cosine decay learning rate with 0.03 warmup ratio. We trained all linear layers in the vision backend using a peak learning rate of 1e-4, and FP32 precision. The classifier is obtained using prompts such as "a photo of a {class}."

**Results**: As shown in Table 4, GoRA outperforms baseline methods across all seven image classification tasks. Specifically, GoRA outperforms full fine-tuning by a margin of 1.01; outperforms LoRA-GA by 0.96 and outperforms LoRA-Pro by 0.27. These results demonstrate that GoRA exhibits superior performance across different models and modalities.

## 5 Discussions

In this section, we present a comprehensive set of ablation studies to evaluate the effectiveness of GoRA's adaptive rank allocation and initialization strategy. We also examine the impact of GoRA's hyperparameters and discuss their role in shaping performance. Additionally, we explore hyperparameter auto-tuning strategies to improve usability.

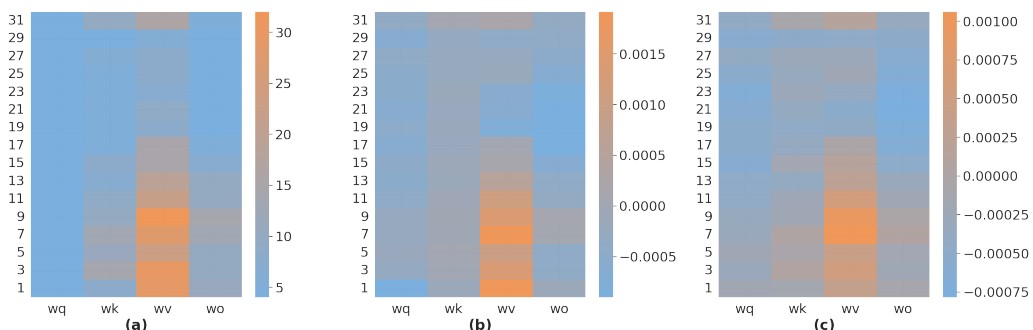

Figure 3: (a) Result rank distribution of fine-tuning Llama-3.1-8B-Base on the MetaMathQA-100K dataset using GoRA;(b) Difference values between GoRA and LoRA in directional updates of pre-trained weights after merging;(c) Difference values between GoRA and LoRA in magnitude updates of pre-trained weights after merging. Data points are presented for every two layers.

## 5.1 The Effect of the Rank Allocation Strategy

The rank allocation strategy is a crucial component that influences the performance of GoRA. As highlighted in Table 5, we conducted ablation studies to evaluate different rank allocation ranges. The results demonstrate that a broader rank allocation range consistently leads to superior performance. For instance, given $\gamma = 5e - 2$, ($r^{\min} = 4, r^{\max} = 32$) achieved a score of 48.98 on HumanEval, significantly outperforming both the fixed rank allocation strategy ($r^{\min} = 8, r^{\max} = 8$) and the more conservative allocation strategy ($r^{\min} = 6, r^{\max} = 15$).

Figure 3 illustrates the rank distribution of ($r^{\min} = 4, r^{\max} = 32$). Notably, most ranks are allocated to the wv layers, while the wq layers receive the fewest rank allocations. This observation aligns

with findings reported in prior work [8]. Moreover, weights with higher ranks receive larger updates after merging the low-rank matrices. These observations underscore the effectiveness of our rank allocation strategy.

Table 5: Ablation study on hyperparameters. To maintain an approximately constant number of trainable parameters, the rank allocation upper bound was reduced as the lower bound was increased.

| Method | $r^{\min}$ | $r^{\max}$ | $\gamma$ | GSM8k | HumanEval |
|--------|-----------|-----------|----------|-------|-----------|
| AdaLoRA | 0 | 12 | - | 70.63±0.77 | 41.46±3.66 |
| LoRA | 8 | 8 | 0 | 67.78±1.25 | 43.09±0.35 |
| GoRA | 4 | 32 | 8e-2 | **72.91±0.76** | 46.54±1.54 |
| GoRA | 4 | 32 | 5e-2 | 72.88±0.99 | **48.98±2.14** |
| GoRA | 4 | 32 | 3e-2 | 72.71±1.22 | 45.93±1.27 |
| GoRA | 4 | 32 | 0 | 72.45±1.14 | 46.34±0.61 |
| GoRA | 0 | ∞ | 5e-2 | 72.83±0.80 | 46.13±3.36 |
| GoRA | 4 | 32 | 5e-2 | 72.88±0.99 | **48.98±2.14** |
| GoRA | 6 | 15 | 5e-2 | 72.25±0.27 | 45.85±3.18 |
| GoRA | 8 | 8 | 5e-2 | 72.10±1.12 | 44.75±3.97 |

## 5.2 The Effect of the Initialization Strategy

Table 5 also summarizes the results of ablation studies conducted with various scaling factors. Our experiments revealed that the choice of scaling factor $\gamma$ has a substantial impact on performance. Notably, GoRA achieves the best performance on HumanEval with $\gamma = 5e - 2$, attaining a score of 48.98. Meanwhile, GoRA with $\gamma = 8e - 2$ slightly outperformed other configurations on the GSM8k, achieving a score of 72.91. Conversely, when $\gamma = 0$, GoRA exhibited the weakest performance on GSM8k, scoring 72.45. A carefully selected scaling factor ensures that the initial low-rank adapter computation closely approximates a gradient descent step, establishing a robust foundation for subsequent optimization. This is crucial for maintaining training stability.

## 5.3 The Effect of Different Importance Metrics

Table 6 compares several importance metrics: parameter sensitivity to loss (the metric used by GoRA), the nuclear norm of the gradient, and the nuclear norm of the parameter–gradient product. The results demonstrate that parameter sensitivity consistently outperforms the other metrics on both GSM8k and HumanEval. Notably, on HumanEval, parameter sensitivity

Table 6: Ablation studies on different importance metrics, where $|| \cdot ||_*$ represents the nuclear norm.

| Metric | GSM8k | HumanEval |
|--------|-------|-----------|
| avg($|\mathbf{W} \odot \mathbf{G}|$) | **72.88±0.99** | **48.98±2.14** |
| $||\mathbf{G}||_*$ | 72.70±0.68 | 43.09±0.93 |
| $||\mathbf{W} \odot \mathbf{G}||_*$ | 72.65±0.78 | 45.12±3.17 |

achieves a score of 48.98, surpassing the nuclear norm of the gradient (43.09) and the nuclear norm of the parameter–gradient product (45.12).

## 5.4 Hyperparameter Auto-tuning

Compared to vanilla LoRA, GoRA introduces two additional hyperparameters: the number of gradient accumulation steps $N$ and the initialization scaling factor $\gamma$. While hyperparameter tuning strategies for LoRA have been well established in prior work, we aim to minimize the additional tuning burden introduced by GoRA to ensure its practical usability. To this end, we design lightweight, automated strategies that effectively eliminate the need for manual tuning of these new parameters:

(1) Adaptive Gradient Accumulation: During gradient accumulation, we monitor the normalized layer-wise importance scores at each accumulation step. Once the change between consecutive importance score sets falls below a small pre-defined threshold (e.g., 0.01), indicating convergence of layer importance, we terminate accumulation early and proceed to the parameter update. This adaptive stopping criterion automatically determines an effective $N$ on the fly, removing the need for manual specification and often accelerating training.

(2) Adaptive Scaling Factor: We hypothesize that the optimal scaling factor $\gamma$ is the one that minimizes the loss of the first training step. To find it, we start from a relatively large initial value (e.g., 1.0) and iteratively reduce it by multiplying by a decay factor (e.g., 0.9) until a small lower bound (e.g., 5e-5)

is reached. For each candidate $\gamma$, we evaluate the loss on the first training batch without performing backpropagation. The $\gamma$ yielding the lowest loss is then used to initialize formal training.

Importantly, the pre-defined constants used in these strategies—such as the convergence threshold (0.01), decay factor (0.9), and lower bound (5e-5) are empirically stable across tasks and model scales. In practice, they require little to no tuning, making GoRA highly usable out of the box while preserving its enhanced representational capacity.

To validate the effectiveness and practicality of our auto-tuning strategies, we conduct ablation studies on the scaling factor $\gamma$ and the gradient accumulation steps $N$. As shown in Table 7, our adaptive methods achieve performance comparable to or better than manually tuned baselines, while eliminating the need for hyperparameter search. Notably, the auto-selected $\gamma$

Table 7: Performance comparison of GoRA with adaptive hyperparameters.

| Method | GSM8k | HumanEval |
|---|---|---|
| Original | 72.91±0.76 | 48.98±2.14 |
| Adaptive $N$ | 72.96±0.19 | 46.85±2.11 |
| Adaptive $\gamma$ | 72.50±0.38 | 50.00±3.23 |

values (0.0549 for CodeFeedback and 0.0858 for MetaMathQA) closely align with those reported in Table 5 (5e-2 and 8e-2, respectively), confirming the reliability of our selection criterion.

## 6 Computational and Memory Analysis

Table 8: Comparison of trainable parameter counts. For GoRA, we set $r^{\min} = 4$ and $r^{\max} = 32$

| Model | Dataset | Target Modules | LoRA | AdaLoRA | GoRA |
|---|---|---|---|---|---|
| T5-Base | SST-2 | all-linear | 3.24M | 4.86M | 3.05M |
| Llama3.1-8B-Base | MetamathQA | attention | 6.82M | 10.24M | 7.00M |
| Llama2-7B-Base | MetamathQA | all-linear | 19.99M | 29.99M | 20.18M |
| CLIP-ViT-B/16 | Cars | vision_model | 1.33M | 2.03M | 1.35M |

During training, GoRA's computational and memory overhead is nearly identical to LoRA's, as both methods maintain a similar number of trainable parameters given a reference rank. Table 8 compares the trainable parameter counts of LoRA, AdaLoRA, and GoRA across various models, with all methods using a rank or average/reference rank of 8.

Table 9: Comparison of time and GPU memory costs during GoRA's initialization process and training process

| Model Size | Process | Time Cost | Memory Cost |
|---|---|---|---|
| 8B | Initialization | 23.60s | 56,139MB |
|  | Training | 37min54.22s | 73,295MB |
| 32B | Initialization | 2min44.65s | 64,821 MB |
|  | Training | 197min52.15s | 66,381 MB |

As outlined in Algorithm 1, GoRA requires a brief gradient pre-computing phase for rank allocation and initialization. However, this overhead is minimal in both computation and memory usage. First, gradient accumulation is performed over only a small number of micro-batches without any optimizer updates. Second, gradients are computed layer-by-layer and immediately offloaded to CPU memory, avoiding the storage of optimizer states. Moreover, Algorithm 2 integrates seamlessly with distributed data Parallelism and can be readily extended to more complex parallel training setups. Specifically, rather than having each worker independently compute and transfer gradients to CPU memory, which would cause massive concurrent PCIe traffic and excessive CPU memory consumption, our method performs an immediate `Reduce` operation on per-layer gradients directly to rank 0 during the backward pass. This ensures that only a single copy of the globally averaged gradient is ever transferred to CPU memory, drastically reducing PCIe bandwidth usage and CPU memory footprint.

To empirically validate the efficiency of GoRA, we measure both training time and memory consumption when fine-tuning Llama-3.1-8B-Base and Qwen2.5-32B [54] on MetaMathQA using 8×A800 80GB GPUs. For Llama-3.1-8B-Base, we use a per-GPU batch size of 8 (without activation checkpoint); for the larger Qwen2.5-32B model, we reduce the per-GPU batch size to 1 (with activation checkpoint [55]) due to memory constraints. We employ the adaptive $N$ strategy described in Section 5.4, and all other experimental settings follow those of our main experiments. As shown in Table 9 (reported memories benefit from Liger kernel [56] and FlashAttention [57]), GoRA incurs negligible additional time cost and no extra memory overhead compared to standard LoRA, even for models of vastly different scales, demonstrating its practicality for real-world deployment.

# 7 Acknowledgement

This paper is supported by National Key Research and Development Program of China (Grants No.2023YFD2000101), National Natural Science Foundation of China (Grants No.32471988, 32271981), and the HFIPS Director's Fund (Grant No.YZJJKX202401).

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

# A  Notations

In this section, we summarize the notations used in the paper in Table 10.

Table 10: List of Notations used in the paper

| Symbol | Description |
|---|---|
| $\mathbf{W} \in \mathbb{R}^{m \times n}$ | Full-rank weight matrix of a linear layer. |
| $\mathbf{W}_0 \in \mathbb{R}^{m \times n}$ | Pre-trained weight matrix of a linear layer in a pre-trained model. |
| $\Delta\mathbf{W} \in \mathbb{R}^{m \times n}$ | Update of the pre-trained weight matrix after fine-tuning. |
| $\Delta\mathbf{W}_t \in \mathbb{R}^{m \times n}$ | Update of the pre-trained weight matrix at fine-tuning step $t$. |
| $\mathbf{W}_t \in \mathbb{R}^{m \times n}$ | Weight matrix of a linear layer at training step $t$ ($\mathbf{W}_t = \mathbf{W}_0 + \Delta\mathbf{W}_t$). |
| $\frac{\partial \mathcal{L}_t}{\partial \mathbf{W}_0} \in \mathbb{R}^{m \times n}$ | Gradient matrix of the pre-trained weight $\mathbf{W}_0$ at step $t$. |
| $\mathbf{G} \in \mathbb{R}^{m \times n}$ | N-batch accumulated gradient matrix of the pre-trained weight $\mathbf{W}_0$. |
| $\mathbf{A} \in \mathbb{R}^{m \times r}, \mathbf{B} \in \mathbb{R}^{r \times n}$ | Trainable low-rank matrices of a LoRA adapter. |
| $\mathbf{A}_0 \in \mathbb{R}^{m \times r}, \mathbf{B}_0 \in \mathbb{R}^{r \times n}$ | Initial values of the low-rank matrices $\mathbf{A}$ and $\mathbf{B}$. |
| $\mathbf{A}_t \in \mathbb{R}^{m \times r}, \mathbf{B}_t \in \mathbb{R}^{r \times n}$ | Trainable low-rank matrices of a LoRA adapter at step $t$. |
| $m$ | Input dimension of the matrix $\mathbf{W}$. |
| $n$ | Output dimension of the matrix $\mathbf{W}$. |
| $r$ | Rank of the low-rank adapter, with $r \ll \min(m, n)$. |
| $r^{\mathrm{max}}$ | Pre-defined maximum rank of a GoRA adapter. |
| $r^{\mathrm{min}}$ | Pre-defined minimum rank of a GoRA adapter. |
| $r^{\mathrm{ref}}$ | Reference rank of GoRA. |
| $\alpha$ | Hyperparameter of LoRA and most of its variants. |
| $\gamma$ | Scaling hyperparameter of GoRA. |
| $\xi$ | Scaling factor of GoRA, automatically determined by $\gamma$. |
| $\mathcal{U}_{\mathrm{kaiming}}$ | Kaiming uniform distribution. |
| $\eta$ | Learning rate used in the optimizer (e.g., AdamW). |
| $\odot$ | Hadamard (element-wise) product of two matrices. |
| $[\cdot]$ | Rounding to the nearest integer. |
| $\|\cdot\|_F$ | Frobenius norm of a matrix. |
| $\|\cdot\|_*$ | Nuclear norm of a matrix. |
| $\mathrm{avg}(\cdot)$ | Average operation (element-wise for a matrix). |

# B  Proofs

## B.1  Proof of optimal approximation of G given A.

Let $\mathbf{G}$ be an $m \times n$ matrix, and $\mathbf{A}$ be an $m \times r$ matrix where $r \ll \min(m, n)$. We aim to derive the projection formula that minimizes the Frobenius norm of the error $\|\mathbf{G} - \hat{\mathbf{G}}\|_F$, where $\hat{\mathbf{G}}$ is the optimal approximation of $\mathbf{G}$ in the column space of $\mathbf{A}$, denoted as $\mathrm{Col}(\mathbf{A})$.

The best approximation $\hat{\mathbf{G}}$ lies in $\mathrm{Col}(\mathbf{A})$, so we can express $\hat{\mathbf{G}}$ as:

$$\hat{\mathbf{G}} = \mathbf{A}\mathbf{B},$$

where $\mathbf{B}$ is an $r \times n$ matrix of coefficients to be determined. Our goal is to find $\mathbf{B}$ such that the error $\|\mathbf{G} - \hat{\mathbf{G}}\|_F$ is minimized.

The error matrix is given by:

$$\mathbf{E} = \mathbf{G} - \hat{\mathbf{G}} = \mathbf{G} - \mathbf{A}\mathbf{B}.$$

To minimize $\|\mathbf{E}\|_F^2$, we take the derivative of $\|\mathbf{E}\|_F^2$ with respect to $\mathbf{B}$ and set it to zero. Expanding $\|\mathbf{E}\|_F^2$, we have:

$$\|\mathbf{E}\|_F^2 = \mathrm{Tr}\left((\mathbf{G} - \mathbf{A}\mathbf{B})^\top (\mathbf{G} - \mathbf{A}\mathbf{B})\right),$$

where Tr represents the trace of a matrix.

Expanding this expression:

$$\|\mathbf{E}\|_F^2 = \mathrm{Tr}(\mathbf{G}^\top \mathbf{G}) - 2\mathrm{Tr}(\mathbf{B}^\top \mathbf{A}^\top \mathbf{G}) + \mathrm{Tr}(\mathbf{B}^\top \mathbf{A}^\top \mathbf{A}\mathbf{B}).$$

Taking the derivative with respect to $\mathbf{B}$ and setting it to zero:

$$-2\mathbf{A}^\top\mathbf{G} + 2\mathbf{A}^\top\mathbf{A}\mathbf{B} = 0.$$

Simplifying:

$$\mathbf{A}^\top\mathbf{A}\mathbf{B} = \mathbf{A}^\top\mathbf{G}.$$

Assuming $\mathbf{A}^\top\mathbf{A}$ is invertible, we solve for $\mathbf{B}$:

$$\mathbf{B} = (\mathbf{A}^\top\mathbf{A})^{-1}\mathbf{A}^\top\mathbf{G}.$$

Substituting $\mathbf{B}$ into $\hat{\mathbf{G}} = \mathbf{A}\mathbf{B}$, we get:

$$\hat{\mathbf{G}} = \mathbf{A}(\mathbf{A}^\top\mathbf{A})^{-1}\mathbf{A}^\top\mathbf{G}.$$

Thus, the best approximation $\hat{\mathbf{G}}$ is:

$$\hat{\mathbf{G}} = \mathbf{A}(\mathbf{A}^\top\mathbf{A})^{-1}\mathbf{A}^\top\mathbf{G}.$$

The matrix $\hat{\mathbf{G}} = \mathbf{A}(\mathbf{A}^\top\mathbf{A})^{-1}\mathbf{A}^\top\mathbf{G}$ is the projection of $\mathbf{G}$ onto the column space of $\mathbf{A}$, and it minimizes the Frobenius norm of the error $\|\mathbf{G} - \hat{\mathbf{G}}\|_F$.

## B.2 Proof of Expectation of Frobenius Norm of $\mathbf{AB}$.

Let $\mathbf{A}$ be a random Gaussian matrix of size $m \times r$, where each element of $\mathbf{A}$ is sampled independently from $\mathcal{N}(0, 1)$. Let $\mathbf{G}$ be a random Gaussian matrix of size $m \times n$, where each element of $\mathbf{G}$ is also sampled independently from $\mathcal{N}(0, 1)$. Define:

$$\mathbf{B} = (\mathbf{A}^\top\mathbf{A})^{-1}\mathbf{A}^\top\mathbf{G},$$

and consider the product:

$$\mathbf{AB} = \mathbf{A}(\mathbf{A}^\top\mathbf{A})^{-1}\mathbf{A}^\top\mathbf{G}.$$

The goal is to compute the expected Frobenius norm $\mathbb{E}[\|\mathbf{AB}\|_F]$, where the Frobenius norm is defined as:

$$\|\mathbf{AB}\|_F = \sqrt{\sum_{i,j}(\mathbf{AB})_{ij}^2}.$$

First, observe that $\mathbf{AB}$ can be rewritten as:

$$\mathbf{AB} = \mathbf{A}(\mathbf{A}^\top\mathbf{A})^{-1}\mathbf{A}^\top\mathbf{G}.$$

Let $\mathbf{P} = \mathbf{A}(\mathbf{A}^\top\mathbf{A})^{-1}\mathbf{A}^\top$. Note that $\mathbf{P}$ is a projection matrix onto the column space of $\mathbf{A}$, and thus $\mathbf{P}$ satisfies:

$$\mathbf{P}^2 = \mathbf{P}, \quad \mathbf{P}^\top = \mathbf{P}, \quad \text{and} \quad \text{rank}(\mathbf{P}) = r.$$

Substituting $\mathbf{P}$ into the expression for $\mathbf{AB}$, we have:

$$\mathbf{AB} = \mathbf{PG}.$$

The Frobenius norm of $\mathbf{AB}$ is given by:

$$\|\mathbf{AB}\|_F^2 = \|\mathbf{PG}\|_F^2 = \text{Tr}((\mathbf{PG})(\mathbf{PG})^\top).$$

Since $(\mathbf{PG})^\top = \mathbf{G}^\top\mathbf{P}$, this becomes:

$$\|\mathbf{AB}\|_F^2 = \text{Tr}(\mathbf{PGG}^\top\mathbf{P}).$$

The matrix $\mathbf{GG}^\top$ is an $m \times m$ random Wishart matrix. When $\mathbf{G}$ is a standard Gaussian matrix of size $m \times n$, the expected value of $\mathbf{GG}^\top$ is:

$$\mathbb{E}[\mathbf{GG}^\top] = n \cdot \mathbf{I}_m,$$

where $\mathbf{I}_m$ is the $m \times m$ identity matrix. Substituting this result into the expression for $\|\mathbf{AB}\|_F^2$, we get:

$$\mathbb{E}[\|\mathbf{AB}\|_F^2] = \mathbb{E}[\text{Tr}(\mathbf{PGG}^\top \mathbf{P})] = \text{Tr}(\mathbf{P}\mathbb{E}[\mathbf{GG}^\top]\mathbf{P}).$$

Using $\mathbb{E}[\mathbf{GG}^\top] = n \cdot \mathbf{I}_m$, this simplifies to:

$$\mathbb{E}[\|\mathbf{AB}\|_F^2] = \text{Tr}(\mathbf{P}(n \cdot \mathbf{I}_m)\mathbf{P}) = n \cdot \text{Tr}(\mathbf{P}^2).$$

Since $\mathbf{P}^2 = \mathbf{P}$, we have:

$$\mathbb{E}[\|\mathbf{AB}\|_F^2] = n \cdot \text{Tr}(\mathbf{P}).$$

The trace of $\mathbf{P}$ is equal to its rank, which is the dimension of the column space of $\mathbf{A}$. Since $\mathbf{A}$ is a $m \times r$ matrix, we have:

$$\text{Tr}(\mathbf{P}) = r.$$

Thus:

$$\mathbb{E}[\|\mathbf{AB}\|_F^2] = n \cdot r.$$

Taking the square root, the expected Frobenius norm of $\mathbf{AB}$ is:

$$\mathbb{E}[\|\mathbf{AB}\|_F] = \sqrt{n \cdot r}.$$

## C   Implementation Details

### C.1   Baseline Methods

We compared GoRA with baseline methods to demonstrate the effectiveness of our approach:

a. **Full**: Trains all parameters in the target layers, resulting in the highest memory consumption.

b. **LoRA** [8]: Introduces low-rank adapters into the target layers, significantly reducing the number of trainable parameters.

c. **Convergence Optimization Methods for LoRA**
   - **rsLoRA** [11]: Modifies the scaling factor in LoRA from $\frac{\alpha}{r}$ to $\frac{\alpha}{\sqrt{r}}$, enabling better performance with higher-rank adapters and stabilizing the training processes.
   - **DoRA** [45]: Decomposes the weight updates of pre-trained weights into magnitude and direction components, and applies LoRA to update only the direction.
   - **LoRA+** [46]: Addresses the imbalance between matrices $\mathbf{A}$ and $\mathbf{B}$ in LoRA by assigning a relatively larger learning rate to matrix $\mathbf{B}$ than to matrix $\mathbf{A}$.

d. **Initialization Optimization Methods for LoRA**
   - **OLoRA** [27]: Initializes LoRA weights using the QR decomposition of the corresponding pre-trained weights.
   - **PiSSA** [16]: Initializes LoRA weights based on the dominant singular vectors obtained from the SVD of pre-trained weights.
   - **LoRA-GA** [18]: Initializes LoRA weights using significant singular vectors derived from the SVD of gradients of pre-trained weights.

e. **Adaptive Methods for LoRA**
   - **AdaLoRA** [14]: Approximates the low-rank adapter structure using SVD, enabling dynamic rank allocation through singular value masking. It also introduces an orthogonal regularization term to the loss function for enhancing orthogonality among features in the low-rank adapter.

### C.2   Implementation Details for Baseline Methods

Several baseline methods introduce tunable hyperparameters compared with vanilla LoRA [8]. To ensure a fair comparison, we adopt the optimal settings reported in the original papers whenever possible. Specifically, for LoRA+ [46], we set the learning rate ratio of matrices A and B to 16. For LoRA-GA [18], we use the "stable" scaling method (the scaling hyperparameter $\gamma$ of LoRA-GA is configured to 16) and manipulate the pre-trained weights during initialization. For AdaLoRA [14], the initial rank is set to 12, the final rank to 8, with $t_i = 150$ and $t_f = 900$. For PiSSA [16], the number of iterations for fast SVD is set to 64.

## C.3 Implementation Details for GoRA

For all experiments with $r^{ref} = 8$, except for the model trained on MetaMathQA [47], we set the scaling factor $\gamma$ to $5e - 2$. For the model trained on MetaMathQA, $\gamma$ is set to $8e - 2$. For all experiments with $r^{ref} = 32$, the scaling factor is set to $1e - 2$; and for $r^{ref} = 128$, we set the scaling factor to $5e - 3$. This is because we observe that more gradient information is compressed by GoRA's initialization with a higher rank, even if $\gamma$ can control the magnitude of the initialization results. To address the imbalance in GoRA's matrices $\mathbf{A}$ and $\mathbf{B}$, we set the learning rate of matrix $\mathbf{B}$ to be 16 times that of matrix $\mathbf{A}$ Throughout the experiments, the $r_{max}$ was empirically defined as $4 \times r^{ref}$, the $r_{min}$ was set to $r^{ref}/2$ (as this setting can maintain a comparable parameter count compared to LoRA), and the gradient accumulation step for GoRA's initialization was set to 64. In the ablation studies, we adhered to the same hyperparameter settings as in the main experiments, unless otherwise specified.

Furthermore, Algorithm 1 presents a basic algorithm for non-parallel scenarios; however, parallel training strategies, particularly data parallelism, are widely employed for training large language models. We introduce the GoRA algorithm under distributed data parallelism in Algorithm 2, which is readily integrable into more complex parallel frameworks.

---

**Algorithm 2** Rank Allocation and Initialization of GoRA under Distributed Data Parallelism

---

**Input:** Number of layers $L$, gradient accumulation steps $N$, model parameters $\theta = \{W_0^l\}_{l=1}^{L}$, current data parallel worker ID $w$, total data parallel workers $W$
**Output:** Initialized low-rank matrices $\{\mathbf{A}_0^l\}_{l=1}^{L}$, $\{\mathbf{B}_0^l\}_{l=1}^{L}$
 1: Initialize empty CPU buffers $\{\mathbf{G}_{\text{avg}}^l\}_{l=1}^{L}$ only on **worker 0**
 2: **for** $i = 1$ to $N$ **do**
 3:     Sample mini-batch and compute loss $\mathcal{L}$
 4:     **for** $l = 1$ to $L$ **do**
 5:         Compute local gradient on GPU: $\mathbf{G}^l \leftarrow \frac{\partial \mathcal{L}}{\partial \mathbf{W}_0^l}$
 6:         **if** $w = 0$ **then**
 7:             Accumulate on CPU: $\mathbf{G}_{\text{avg}}^l \leftarrow \mathbf{G}_{\text{avg}}^l + \frac{1}{N}\mathbf{G}^l$
 8:         Release the GPU memory occupied by $\mathbf{G}_l$: $\mathbf{G}_l \leftarrow$ None
 9: **if** $w = 0$ **then**
10:     **for** $l = 1$ to $L$ **do**
11:         Compute the importance $I(\mathbf{W}_0^l)$ as detailed in Algorithm 1
12:     Broadcast the importance set $\{I(\mathbf{W}_0^l)\}_{l=1}^{L}$ to all other workers
13: **else**
14:     Receive $\{I(\mathbf{W}_0^l)\}_{l=1}^{L}$ from worker 0
15: **for** $l = 1$ to $L$ **do**
16:     **if** $w = 0$ **then**
17:         Load the gradient $\mathbf{G}_{avg}^l$ from CPU to GPU
18:         Initialize $\mathbf{A}_0^l, \mathbf{B}_0^l$ as detailed in Algorithm 1
19:         Clear the GPU memory occupied by $\mathbf{G}_{\text{avg}}^l$
20:         Broadcast initialized $\mathbf{A}_0^l, \mathbf{B}_0^l$ to all other workers
21:     **else**
22:         Receive initialized $\mathbf{A}_0^l, \mathbf{B}_0^l$ from worker 0
23: **Return** $\{\mathbf{A}_0^l\}_{l=1}^{L}, \{\mathbf{B}_0^l\}_{l=1}^{L}$

---

## C.4 Hyperparameters for Each Experiment

The hyperparameters used in each experiment are summarized in Table 11. For experiments on T5-Base and Llama2-7B-Base, we adopt the settings from LoRA-GA [18]; for CLIP-ViT-B/16, we follow LoRA-Pro [53]. For experiments on Llama3.1-8B-Base, including both baseline methods and GoRA, we use one of the most commonly adopted hyperparameter configurations.

Table 11: Hyperparameters used in experiments

| Model | LR | LR Decay | Warmup | Optimizer | Betas | Weight Decay | Batch Size |
|---|---|---|---|---|---|---|---|
| T5-Base | 1e-4 | 0 | 0.3 | Adam | 0.9, 0.999 | 0 | 32 |
| Llama3.1-8B-Base | 5e-5 | 0.1 | 0.3 | AdamW | 0.9, 0.999 | 5e-4 | 64 |
| Llama2-7B-Base | 2e-5 | 0 | 0.3 | AdamW | 0.9, 0.999 | 0 | 32 |
| CLIP-ViT-B/16 | 1e-4 | 0 | 0.3 | Adam | 0.9, 0.999 | 0 | 64 |

## C.5 Training Environments

For natural language understanding tasks reported in section 4.1, we conduct our experiments using the Huggingface Transformers framework for model and trainer implementation on a single RTX 4090 24GB. In contrast, for natural language generation tasks reported in Section 4.2 and Section 5, we utilize the DeepSpeed ZeRO2 [25] data parallel framework and FlashAttention-2 [58] mechanism, leveraging the power of 8 RTX 4090 24 GB GPUs or 8 A800 80 GB GPUs. All codes of GoRA and baseline methods are implemented in PyTorch.

# D   Addtional Experiments

## D.1   Computational Overhead of Pseudo-Inverse Initialization

The initialization strategy of GoRA involves computing a pseudo-inverse for each low-rank adapter matrix. Although the theoretical cost of pseudo-inversion scales cubically with rank ($\mathcal{O}(r^3)$), our implementation performs all operations on GPUs using optimized linear algebra routines, resulting in minimal practical overhead.

We benchmark the total initialization time for the Llama-3.1-8B-Base model across different ranks following the settings of our main experiments. Results are summarized in Table 12. Even at rank 128, where each pseudo-inverse requires approximately 2 million FLOPs, the entire initialization completes in under 4 seconds. This amounts to less than 0.1% of typical fine-tuning runtime, confirming that the pseudo-inverse step does not constitute a computational bottleneck in practice.

Table 12: Pseudo-inverse initialization time for Llama-3-8B-Base (GPU, end-to-end).

| Rank | Initialization Time |
|---|---|
| 8 | 1.40s |
| 32 | 1.56s |
| 128 | 3.43s |

## D.2   Combining GoRA with QLoRA

To investigate whether GoRA can be effectively integrated with quantization techniques, we adopt the quantization method from QLoRA, which quantizes pre-trained weights to NF4 precision for storage and dequantizes them back to BF16 precision during computation. We evaluate this combined approach, which we term QGoRA, by fine-tuning the Llama-3.1-8B-Base model and assessing its performance on math and code benchmarks, following the same protocol as in our main experiments. As shown in Table 13, GoRA integrates seamlessly with QLoRA's quantization framework and consistently outperforms QLoRA across evaluated tasks.

Table 13: Performance of QGoRA and QLoRA.

| Method | GSM8k | HumanEval |
|---|---|---|
| QLoRA | $65.10 \pm 0.95$ | $43.49 \pm 0.70$ |
| QGoRA | $70.58 \pm 1.33$ | $44.10 \pm 3.68$ |

## D.3   Gradient Reconstruction Accuracy of GoRA Initialization

To assess how well the GoRA initialization approximates the pre-computed gradients, we measure the reconstruction error between the initialized low-rank adapters and the full gradients across all adapted layers in the Llama-3.1-8B-Base model. Results on two datasets are summarized in Table 14. These results suggest that the initialization preserves a high proportion of gradient information (approximately 88–89% in relative terms).

Table 14: Gradient reconstruction error of GoRA initialization.

| Dataset | Absolute Error | Relative Error |
|---|---|---|
| MetaMathQA | 0.0382 | 0.1147 |
| Code-Feedback | 0.0322 | 0.1152 |

## D.4 Multi-Step Gradient Initialization

We explored an alternative initialization variant, GoRA-pro, which estimates layer importance using multi-step stochastic gradients. In this approach, a lightweight pre-training phase performs $n$ exploratory updates per layer using the AdaLomo [59] optimizer; gradients are accumulated on CPU and then discarded, and the original weights are restored before adapter initialization. This process uses approximately 23% less GPU memory than full fine-tuning (56.2 GB vs. 73.0 GB for Llama-3.1-8B-Base).

We evaluate GoRA-pro on GSM8K and HumanEval, comparing it to the standard GoRA (which uses one-step gradient accumulation). Results are shown in Table 15. The comparable performance suggests that the multi-step gradient idea is viable and may offer further improvements with careful tuning of the exploration schedule, gradient normalization. We leave a systematic investigation of this direction to future work.

Table 15: Comparison of GoRA and GoRA-pro (Llama-3-8B-Base).

| Method | GSM8K | HumanEval |
|---|---|---|
| GoRA | $72.91 \pm 0.76$ | $48.98 \pm 2.14$ |
| GoRA-pro | $72.30 \pm 0.30$ | $47.36 \pm 1.54$ |

## E Clarifications

### E.1 Clarification on the training-inference gap introduced by previous initialization methods

This is due to their reliance on manipulating pre-trained weights. Specifically:

**Manipulation of Pre-Trained Weights**: These methods are required to manipulate the value of pre-trained weights during initialization, as $\mathbf{A}_0\mathbf{B}_0 \neq 0$ and $\mathbf{W}_0 + \mathbf{A}_0\mathbf{B}_0 \neq \mathbf{W}_0$. As a result, during inference, the term $\mathbf{A}_0\mathbf{B}_0$ must be recomputed to properly reconstruct the adapted weight matrix for effective model deployment, which is essential for correct model outputs.

**The Inconvenience of Recalculating Initialization Results**: During inference, it is often infeasible to recompute the initialization results for methods that either rely on randomness, such as PiSSA[16], or require access to training data, such as LoRA-GA [18] and EVA[28]. Furthermore, for approaches like MiLoRA [17], the initialization process itself can be computationally expensive and time-consuming.

**Incompatibility Across Multiple Adapters**: When multiple adapters are trained on different tasks using previous data-driven non-zero initialization methods, the pre-trained weights are manipulated inconsistently. As the result of $\mathbf{A}_0\mathbf{B}_0$ depends on the task. This makes it challenging to serve multiple adapters simultaneously, limiting flexibility in multi-task scenarios.

**Saving manipulated pre-trained weights sacrifices one of the key advantages of LoRA**: While it is possible to merge the low-rank adapter weights into the pre-trained weights after training, saving the pre-trained weights post-merging sacrifices one of the key advantages of LoRA: minimal storage requirements (e.g., 10MB compared to 14GB). Other potential approaches to eliminate the training-inference gap and their limitations are discussed in Section 2.2.

### E.2 Compare GoRA's initialization strategy with LoRA-GA

Both GoRA and LoRA-GA leverage gradients to initialize low-rank adapters, but there are several key differences:

1. **Motivation**:
   - LoRA-GA: Minimizes the difference in updates of weights between LoRA and full fine-tuning.

- GoRA: Views LoRA adapters, gradient compressors, and optimizes the compression form.

2. **Scaling factor**:
   - LoRA-GA: Inspired by rsLoRA, its scaling aims to stabilize training.
   - GoRA: Scale the initialized product of adapters to any desired magnitude flexibly.

3. **Methodology**:
   - LoRA-GA: Initialize the weights using SVD decomposed gradients.
   - GoRA: Initialize the weights of $\mathbf{B}$ using gradients compressed by pseudo-inverse of random initialized $\mathbf{A}$.

### E.3   Further compare with previous related works

Building upon previous works, GoRA makes several unique contributions:

1. First data-driven initialization method without manipulating pre-trained weights.

2. Efficient rank allocation strategy without additional trainable parameters and training complexity.

3. Unified framework for gradient-driven rank allocation and initialization.

## F   Limitations And Future Works

In this study, we demonstrate that GoRA outperforms baseline low-rank adaptation methods and achieves performance comparable to full fine-tuning. However, our evaluation has not yet extended to larger models and more extensive datasets. We hypothesize that for larger models, such as Llama-3.1-70B [2], GoRA could more effectively leverage the pre-trained knowledge inherent in these models. Additionally, while this research primarily focuses on language models and natural language processing tasks, there is potential to generalize GoRA to a broader range of model types and tasks, such as visual language models and visual question answering.

Another limitation of this study is that the initialization of matrix $\mathbf{A}$ is not restricted to random initialization. Employing alternative methods, such as extracting distinguishing features from pre-trained weights to initialize the matrix $\mathbf{A}$, could potentially enhance performance, as it would combine the benefits of both experience-driven and data-driven initialization approaches. Furthermore, it is worth noting that GoRA demonstrates theoretical compatibility with other LoRA variants, such as DoRA [45]. These promising avenues remain to be explored in future research endeavors.

