# OpenReview forum: "GoRA: Gradient-driven Adaptive Low Rank Adaptation"
_NeurIPS.cc/2025/Conference — NeurIPS 2025 poster_

### Official Review · Reviewer_bxar · 2025-06-25

**Clarity:** 4
**Significance:** 2
**Originality:** 3
**Rating:** 4
**Confidence:** 3

**Summary:**

This paper presents a novel low-rank adaptation technique that is capable of adaptive rank selection as well as informative initialization all within the same theoretical framework. The core idea of the proposed method is based on the importance accumulated gradients from the original backpropagation without LoRA for guiding the architecture and the initialization for LoRA. The authors have shown the empirical merits of the proposed by performing extensive experimentation as well as ablation studies to assess the importance of each component.

**Questions:**

- Does the memory usage ion Table 8 also include the gradient accumulation step?
- The authors claimed that the gradient accumulation step doesn't lead to extra memory usage. Why is that the case?
- The proclaimed "$n$-step gradient accumulation" is in fact a 1-step gradient accumulation over a big batch. Why this choice and not the actual $n$-step gradient accumulation which is more intuitive? Any empirical comparisons between the two?

**Ethical Concerns:**

["NO or VERY MINOR ethics concerns only"]

**Final Justification:**

I'll maintain my current score as I believe while the paper is above the acceptance threshold, some critical explanations need to be added to the revised draft.

**Limitations:**

The main limitation IMO that hasn't been discussed in the Limitation section is that the initial gradient accumulation step can be prohibitive in terms of the memory usage for larger scale models. This kind of defeats the purpose of using LoRA in the first place. I'd strongly recommend the authors to discuss this issue not just in the Limitation section but in the main body of the paper.

**Paper Formatting Concerns:**

No formatting issues that I observed.

**Quality:**

3

**Strengths And Weaknesses:**

Strengths:
- The theoretical foundations of the proposed methodology is intuitive and well-explained.
- The experimental study is reasonably extensive with promising results.
- The paper is generally very well-written.

Weaknesses:
- The proposed methodology requires to perform $n$-step gradient calculations on the original weights. However, this is prohibitive in many practical situations mainly due to the memory constraints, and hence the use of low-rank adaptation techniques in the first place. The presented experiments are only done for medium size models which are likely to be ok, but there's no indication that'd scale up. The authors have put one sentence in Algorithm 1 claiming that the gradient accumulation step would not lead to extra memory usage. but this claim makes no sense, plus there's no further argument to support this claim. Also, it's not clear the memory usage peaks reported in Table 8 include the  gradient accumulation preprocessing step or just the GORA optimization.

- The proclaimed "$n$-step gradient accumulation" is not really the $n$-step gradient accumulation, is it? In Algorithm 1, it's mentioned that the step function of the optimizer is not applied which makes the whole gradient accumulation step equivalent to a 1-step gradients on a larger batch. There's no explanation whatsoever why this choice was made over the actual n$-step gradient accumulation which seems to be more intuitive.

Minor issues:
- The reference in Algorithm 1 Line 3 is broken.
- Confusing notation: $n$ is used for two different things in the paper.
- The Hadamard product notation is inconsistent between Eq. 5 and Table 6.

---

> ### Author Rebuttal · Authors · 2025-07-31
>
> We sincerely appreciate the reviewer's insightful and encouraging comments regarding our work. We are grateful for their recognition of the theoretical foundations, extensive experimental validation, and clear presentation of our proposed method. Below we provide detailed responses to address the reviewer's specific concerns.
>
> ## Q1. Does the memory usage ion Table 8 also include the gradient accumulation step? Why gradient accumulation step doesn't lead to extra memory usage?
>
> We appreciate the reviewer's attention to the initialization cost of GoRA. We would like to provide further clarification about our initialization algorithm's memory efficiency and scalability.
>
> **Memory Management During Initialization:**
>
> Our initialization algorithm employs a carefully designed memory management scheme:
>
> 1. After computing gradients for a specific layer:
>     - Gradients are immediately averaged across GPUs via all-reduce
>     - Only the local rank-0 process stores the gradients in CPU memory
>     - All other processes discard their gradients
> 2. Memory footprint breakdown:
>     - *GPU memory*:
>         - 16-bit model weights
>         - 16-bit activations
>         - 16-bit **single-layer gradients**
>     - *CPU memory*:
>         - One copy of 16-bit gradients for all target modules (e.g., 2-3GB for attention modules in an 8B model)
>
> **Comparison with Training Phase Memory Usage:**
>
> - *Training phase* GPU memory includes:
>     - 16-bit model weights
>     - 16-bit activations
>     - 16-bit full-model gradients
>     - 32-bit trainable parameter copies
>     - 32-bit optimizer states
> - The peak GPU memory usage during initialization remains consistently lower than during training, as demonstrated in Table 8 (which already accounts for initialization memory). Under the experimental conditions reported in this paper, the initialization process consumes **17.35GB** of memory, while training requires **19.75GB**.
>
> **Empirical Validation on Large Models:**
>
> We conduct an additional experiment on a 32B model under the following conditions:
>
> 1. **Hardware Configuration:**
>     - 8 × NVIDIA A800 80GB GPUs
>     - 105 × Intel Xeon Platinum 8336C CPUs @ 2.30GHz
>     - 1.875TB RAM
> 2. **Software Configuration:**
>     - Model: Qwen2.5-32B-Instruct
>     - Dataset: MetaMathQA-395K
>     - Batch size: 1 per GPU
>     - Gradient accumulation steps: 8
>     - Optimizations: Liger kernel + FlashAttention + Activation Checkpoint
>
> **Key Results:**
>
> 1. *Initialization Phase*:
>     - Completed in 18 steps (early stopped from planned 64 steps)
>     - Total time: **2 minutes 44.65 seconds (~8.4s/step)**
>     - GPU memory: **64,821 MB**
> 2. *Training Phase*:
>     - Total time: ~11 hours
>     - GPU memory: **66,381 MB**
>
> These results demonstrate that GoRA's initialization remains more memory-efficient than training even at 32B scale. For extremely large models (70B+), we can further optimize memory by:
>
> - Offloading model weights to CPU during initialization
> - Reducing GPU memory footprint to just single-layer weights
>
> This design ensures GoRA's scalability while maintaining its theoretical advantages.
>
> ##  Q2. n-step gradient
> We sincerely appreciate this insightful observation. The reviewer is correct—our original method computes gradients over *n*  micro-batches but applies them in a single aggregated step (equivalent to large-batch training), rather than performing sequential*n*-step updates. We apologize for the unclear terminology and have revised the manuscript to explicitly refer to this as **one-step gradient accumulation** to avoid confusion.
>
> **Introducing GoRA-pro: A Novel *n*-Step Gradient Framework**
>
> Motivated by the reviewer's question, we developed **GoRA-pro**, a memory-efficient algorithm that leverages true*n*-step gradients for rank allocation. Here's how it works:
>
> 1.**Lightweight Pre-Training Phase**:
>    - A layer-wise SGD optimizer performs *n* exploratory updates (*η*: learning rate):
>      $$p = p - \eta \cdot \nabla p$$
>    - Gradients are discarded immediately after updates, minimizing memory (only **56,151 MB** vs.**72,971 MB** in formal training).
>
> 2.**Delta-Weight Calculation**:
>    - After*n* steps (~1-2s/step), we compute the *n*-step gradient per layer:
>      ```python
>      p.grad_stored += p.grad.detach().clone().cpu()  # Aggregate gradients
>      p.iters += 1
>      ```
>    - The final gradient is normalized:
>      ```
>      grad_stored = p.grad_stored / p.iters
>      ```
>
> 3.**Rank Allocation & Reset**:
>    - Layer importance is estimated using the*n*-step gradient and a CPU-stored reference model $W_{\text{cpu}}$.
>    - The original weights are restored, and low-rank adapters are initialized with the compressed gradients.
>
> **Advantages & Results**
>
> GoRA-pro bridges the gap between theoretical*n*-step gradients and practical constraints:
>
> -**Accuracy**: Simulates full fine-tuning dynamics more faithfully than single-step methods.
>
> -**Efficiency**: No extra memory overhead (exploratory phase uses 23% less memory than formal training).
>
> -**Novelty**: To our knowledge, this is the first work to use*n*-step gradients for low-rank adapter initialization.
>
> | Method | GSM8K | HumanEval |
> | --- | --- | --- |
> | GoRA | 72.91±0.76 | 48.98±2.14 |
> | GoRA-pro | 72.30±0.3 | 47.36±1.54 |
>
> We will include empirical comparisons (one-step vs.*n*-step) and a broader discussion of this direction in the revised manuscript. Thank you for highlighting this impactful opportunity!
>
> ## Minor issues
> Thank you for reminding! We will fix these issues in the revised version of our paper!
>
> ## Limitations
>
> We appreciate the reviewer's thoughtful comment regarding memory usage during the initial gradient accumulation step. We agree this is an important consideration for practical applications, particularly when scaling to larger models.
>
> As we addressed in our response to Q1, our initialization method does not increase peak memory consumption during training. For clarity, we will:
>
> 1. Expand the discussion in Section 5.4 ("Computation and Memory Analysis") to explicitly address this point
> 2. Include a more detailed comparison of memory requirements between our initialization process and standard LoRA training process
>
> The current brevity of this discussion was due to page constraints rather than oversight. We appreciate the opportunity to make this aspect clearer in our revised manuscript.

---

> > ### Comment · Reviewer_bxar · 2025-08-03
> >
> > I appreciate the authors detailed response to my questions and comments. Please include the detained description of your initialization procedure in the revised manuscript. I also like the introduction of GoRA-pro as it'd add a new angle for further extension of the current framework.

---

> > > ### Author Response · Authors · 2025-08-04
> > > **Thanks to reviewer bxar!**
> > >
> > > Thank you for your positive feedback and constructive suggestions. We will incorporate the detailed description of the initialization procedure in the revised manuscript as requested. We are also pleased that you found the introduction of GoRA-pro valuable for extending the framework. Your insightful comments have significantly improved our paper.

---

### Official Review · Reviewer_stk3 · 2025-06-28

**Clarity:** 4
**Significance:** 3
**Originality:** 4
**Rating:** 5
**Confidence:** 4

**Summary:**

The paper introduces GoRA (Gradient-driven Adaptive Low Rank Adaptation), a novel framework designed to enhance the efficiency and effectiveness of fine-tuning large language models (LLMs) using Low-Rank Adaptation (LoRA). GoRA addresses two critical limitations of existing LoRA variants: static rank selection and suboptimal weight initialization. It proposes a unified approach that leverages gradient information during training to dynamically assign optimal ranks to low-rank adapters and initialize their weights adaptively. The authors present extensive experiments across various architectures and modalities, demonstrating that GoRA consistently outperforms existing LoRA-based methods and can even surpass full fine-tuning in certain challenging tasks, all while maintaining the computational efficiency of vanilla LoRA.

**Questions:**

- Gradient Collection Memory Footprint: The paper states that the initial gradient collection "do not require extra memory as shown in Table ??", but the table reference is missing. Could the authors please provide specific figures for the memory footprint (e.g., in GB) required for gradients when offloaded to CPU for a representative model like Llama3.1-8B-Base? Furthermore, can you elaborate on whether this memory usage remains practically insignificant for models orders of magnitude larger (e.g., 70B+ parameters), where the full W matrix and its corresponding gradients could be very substantial?
- Scalability of Initial Gradient Computation Time: The initial gradient computation takes 4 minutes for the Llama-3.1-8B-Base model . Please provide insights into how this 4-minute overhead scales if the number of gradient accumulation steps in Algorithm 1 were significantly increased, or if the method were applied to models with considerably more parameters than the 8B-parameter Llama models tested. Could this initial phase become a bottleneck in very large-scale fine-tuning?
- Dynamic Initialization of Matrix A: The paper mentions the non-random initialization of matrix A as a direction for future work . Given that the initialization of matrix B depends directly on A , could the authors discuss the theoretical or practical complexities anticipated in developing a data-driven or non-random initialization for A, especially in maintaining the "optimal gradient compression" property and computational efficiency?

**Ethical Concerns:**

["NO or VERY MINOR ethics concerns only"]

**Final Justification:**

I acknowledge that i have read the authors' rebuttal to all reviewers. The response by the authors and the additional details and experiments provided further solidifies my initial assessment of the work. I will keep my original high rating of 5.

**Limitations:**

The authors have adequately addressed the limitations of their work in Appendix D, titled "Limitations And Future Works" . They acknowledge that their evaluation has not yet extended to larger models (like Llama-3.1-70B) and more extensive datasets, and that the primary focus has been on language models and NLP tasks, with potential for generalization to other modalities . They also identify the current restriction of matrix A to random initialization as a future area for improvement  and note theoretical compatibility with other LoRA variants . This transparency about the scope and future work is commendable.

**Paper Formatting Concerns:**

- Grammar (Line 3 in Alg 1): "This operation do not require extra memory" should be "This operation does not require extra memory".
- Typo (Line 235): "provieded"  -> "provided".
- Grammar (Line 126): "require"-> "requires"
- Grammar (Line 130): "but this reduces the their usability." -> "but this reduces their usability."
- Typo (Line 521):: "whcih”  -> "which"
- Grammar (Lines 521 and 523): "which comparable"  -> "which are comparable".

**Quality:**

4

**Strengths And Weaknesses:**

Strengths:
- Originality and Unified Framework: GoRA represents a novel advancement to unify both adaptive rank selection and adaptive initialization strategies within a single framework. This contrasts with prior approaches that typically address these aspects in isolation, leading to a more holistic and effective solution for LoRA fine-tuning.
- Novel Initialization Strategy: GoRA's adaptive initialization strategy for matrix B, utilizing pseudo-inverse compressed gradients, is novel and theoretically motivated. This approach crucially avoids manipulating pre-trained weights, thereby eliminating the "training-inference gap" that affects many existing non-zero initialization methods.
- Effective Adaptive Rank Allocation: The proposed gradient-driven, sensitivity-based importance metric for rank allocation is well-justified through ablation studies, demonstrating superior performance compared to alternative metrics like nuclear norms .
- Superior Performance: The paper provides compelling evidence that GoRA consistently outperforms a wide array of LoRA-based methods across natural language understanding (NLU), natural language generation (NLG), and image classification tasks. Notably, GoRA demonstrates the capability to surpass full fine-tuning performance in challenging tasks such as mathematical reasoning.
- Computational Efficiency Preservation: A key advantage of GoRA is its ability to achieve performance improvements while preserving the efficiency of vanilla LoRA, both in terms of training time and peak GPU memory usage. This is a critical factor for fine-tuning large models under resource constraints. It also maintains a similar number of trainable parameters compared to standard LoRA.
- Comprehensive Experimental Validation: The evaluation covers diverse models (T5, Llama, CLIP-VIP) and modalities (NLP, Vision), showcasing GoRA's broad applicability and robust performance across different contexts. The inclusion of ablation studies provides valuable insights into the contribution of individual components .
- Clarity and Reproducibility: The paper is well-structured and clearly explains the technical contributions, including a detailed algorithm for GoRA's method. The provided experimental settings, hyperparameters, and implementation details enhance the reproducibility of the reported results . Proofs are also included in the appendix. Code is also provided.

Weaknesses:
- Ambiguous Memory Claim for Gradient Collection: The paper states that the initial gradient computation "do not require extra memory as shown in Table ??". However, the table reference is missing, and neither Table 7 (trainable parameters) nor Table 8 (training time, peak GPU memory) explicitly quantifies the memory used for storing the accumulated gradients for all layers, even if offloaded to CPU. While CPU memory might be less constrained than GPU memory, for extremely large models where W matrices can be order of 10s of gigabytes, this unquantified memory footprint for gradient storage could still be a practical concern, making the claim of "no extra memory" problematic without direct evidence.
- Scalability of Initialization Phase for Larger Models/Datasets: Although the initial gradient computation phase takes only 4 minutes for the Llama-3.1-8B-Base experiment , the paper does not extensively discuss how this initial overhead scales with significantly larger models (e.g., Llama-3.1-70B) or much larger datasets, which could potentially make this phase a more substantial part of the total training time.
- Limited Guidance on $\alpha$ Hyperparameter Selection: While the paper extensively detail the selection and impact of $\gamma$ through ablation studies and provide specific values in their experiments, the paper does not explicitly state how $\alpha$ is selected or what its numerical value is in the experiments conducted for GoRA. Can the authors comment on its selection?

---

> ### Author Rebuttal · Authors · 2025-07-31
>
> We sincerely thank the reviewer for their thoughtful and encouraging assessment of our work, particularly their recognition of GoRA’s originality, computational efficiency, and comprehensive validation. We greatly appreciate the time and care taken to highlight the strengths of our framework. Below, we address the reviewer’s concerns point by point.
>
> ## Q1. Memory Claim
> We sincerely appreciate the reviewer's careful examination of the memory issues. First, we apologize for the citation oversight in our paper - the correct reference should be Table 8. Regarding memory consumption:
>
> 1. Our method introduces no additional GPU peak memory:
>     - Initialization peak memory: 16-bit model params + 16-bit activations + single-layer gradients
>     - Training peak memory: 16-bit model params + 32-bit optimizer states + 32-bit param copies + 16-bit activations
>     - Even with ZeRO-2 optimization, initialization memory remains significantly lower than training memory (as shown in Table 8)
>     - Optional: Model can be offloaded to CPU during initialization for further GPU memory reduction
> 2. CPU memory clarification:
>     - Storing 16-bit targeted modules’ gradients does incur additional CPU memory compared to LoRA (the gradients are only saved by the local-rank-0 process)
>     - However:
>
>         a) CPU memory is substantially more affordable than GPU memory
>
>         b) Our experimental platform features 1.8TB CPU memory (per 8-GPU node)
>
>         c) For 100B-parameter models (~200GB), this overhead is entirely manageable
>
>
> We will include detailed memory analysis in the revised version and correct all relevant citations.
>
> ## Q2. Initialization Phase for Larger Models/Datasets
>
> We appreciate the reviewer's valuable feedback regarding the scalability of our initialization phase. In response, we conduct a additional experiment on larger-scale models with latest optimized algorithms (as detailed in our responses to Reviewer 4NWP's Q3-Q4). Below are the experimental results:
>
> **Experimental Setup:**
>
> 1. **Hardware:**
>     - 8 × NVIDIA A800 80GB GPUs
>     - 105 × Intel(R) Xeon(R) Platinum 8336C @ 2.30GHz CPUs
>     - 1.875TB RAM
> 2. **Software:**
>     - Model: Qwen2.5-32B-Instruct
>     - Dataset: MetaMathQA-395K
>     - Batch size: 1 per GPU
>     - Gradient accumulation steps: 8
>     - Optimizations: Liger kernel + FlashAttention + Activation Checkpoint
>
> **Results:**
>
> 1. **Initialization Phase:**
>     - Completed in **18 steps** (early stopped from 64 planned steps)
>     - Total time: **2 minutes 44.65 seconds** (~8.4s/step)
>     - GPU memory usage: **64,821 MB**
> 2. **Training Phase:**
>     - Total time: **~11 hours**
>     - GPU memory usage: **66,381 MB**
>
> These results demonstrate that our optimized initialization remains highly efficient even for models at the 32B scale, with negligible overhead compared to full training. We will include this analysis in the revised manuscript.
>
> ## Q3. How alpha is selected
>
> Our setting for $\alpha$ is consistent with previous works[1,2,3] (i.e, $\alpha=16$)
>
> [1] Wang S, Yu L, Li J. Lora-ga: Low-rank adaptation with gradient approximation[J]. Advances in Neural Information Processing Systems, 2024, 37: 54905-54931.
>
> [2] Wang Z, Liang J, He R, et al. Lora-pro: Are low-rank adapters properly optimized? In: Proceedings of the International Conference on Learning Representations, 2025.
>
> [3] Meng F, Wang Z, Zhang M. Pissa: Principal singular values and singular vectors adaptation of large language models[J]. Advances in Neural Information Processing Systems, 2024, 37: 121038-121072.
>
> ## Q4. Dynamic Initialization of Matrix A
>
> Thank you for your insightful suggestions about initialization strategy of Matrix A.  We conduct follow experiments:
>
> | Method | GSM8K | HumanEval |
> | --- | --- | --- |
> | GoRA (A initialized with gradient principle components) | 72.15±1.29 | 47.15±2.31 |
> | GoRA (A initialized with weight principle components) | 58.73±1.06 | 34.55±3.35 |
> | GoRA (reported in the paper) | 72.91±0.76 | 48.98±2.14 |
>
> The results demonstrate that initialize A with gradient principle components yield comparable performance compared to standard GoRA while initialize B with weight principle components yield downgraded performance. We hypothesis that the noisy from the weight principle components can lead to sub-optimal gradient compression. We will further investigate and discuss these strategies in the revised version of our paper.
>
> ## Q5. Typos
> We sincerely appreciate the reviewer's careful attention to detail in identifying these typographical errors. We have thoroughly reviewed the manuscript and corrected all instances in our revised version.

---

### Official Review · Reviewer_zRT2 · 2025-07-01

**Clarity:** 3
**Significance:** 3
**Originality:** 2
**Rating:** 5
**Confidence:** 4

**Summary:**

This work proposed a new parameter-efficient finetuning method, an upgrade to LoRA. It is also a low-rank update method that keeps the bulk of the pretrained model frozen, but introduces two innovations: a) an adapted initialisation strategy that does not start with a zero-initialised matrix (like in LoRA) that tries to minimise the error in the gradient (by comparing it against the full-finetuning gradient), b) it adapts the rank during training using gradient information. On various benchmarks it outperforms baselines LoRA and other popular methods such as PiSSA and DoRA.

**Questions:**

Q:
What if you rerun the initialisation strategy a couple of times during the training? does the ideal rank allocation change drastically or stay consistent? How far off are the trained matrices from the ones that best compress the full gradients? Does this yield better performances

What happens if you initialise not with L1 optimal compression of the gradients but other metrics like L2  (||W * G ||_2)?

Can and how can this be combined with QLoRA type quantisation? Can this scale to 8

small things:
* typo: "this reduces the their usability"

**Ethical Concerns:**

["NO or VERY MINOR ethics concerns only"]

**Final Justification:**

I thank the authors for their rebuttal that addressed most of my concerns.

I do however, disagree on

> Methods like PiSSA rely on randomized SVD to compute, making it impossible to exactly recover the same during inference.

This is quite feasible by just storing a random seed.
But this is minor, i've upgraded to A.

**Limitations:**

yes

**Quality:**

3

**Strengths And Weaknesses:**

+ the experiments cover both vision and NLP (and therein includes encoder as well as decoder LMs)
+ performance gains look solid and are compared against baselines from other papers
+ experiments include st-deviations
+ some analysis of the method is provided
+ initialisation being that important is interesting for the community


Weaknesses
- before training the gradients for all weights are computing in a full-finetuning like fashion to get the gradients. this requires as much memory as full-finetuning so actually makes is quite prohibitive to scale to large LLMs in the 13B+ range.

- the motivation of using non-zero initialisation is states as "Thus, designing a nonzero initialization method without manipulating pre-trained weights remains an open problem." but it's not clear why this is a problem. As the intro also says, substracting AB from the original W seems fine.

- the pseudo-inverse is used to compute the optimal initialisation for compressing the gradient ... in terms of L2 distance, if I understand correctly. It's not clear if this is indeed optimal. one could also find the best initialisation with L1 distance (e.g. by running a few steps of SGD to find the best initialisation). L2 thus seems a bit out of the blue and it would be nice if this motivation is better argued for.

- a lot of work is applied in correctly initialising the matrices including their ranks. But it's not clear if the ideal rank allocation doesn't change during training.

- GoRA, despite its smart rank allocation strategy has quite varying performances depending on the hyperparamter r_ref. E.g. varying from 5.82 on MT-Bench (the worst in the whole table) to 6.34 (second best in the table), or in a reverse of trends, 52 vs 48.9 on HumanEval. This makes it very hard for practioners to just "use" the method if there's such crucial and sensitive hyperparamters.

---

> ### Author Rebuttal · Authors · 2025-07-31
>
> ## Q1. Gradient Computing Cost
> We appreciate the reviewer's important observation regarding memory requirements during gradient computation. To clarify our approach:
>
> 1. **Memory-Efficient Initialization:**
>     - We employ layer-wise gradient computation with immediate all-reduce and CPU offloading
>     - This eliminates the need to store gradients on GPU
> 2. **Optimizer State Advantage:**
>     - Our method requires no optimizer states during initialization
> 3. **Scalability:**
>     - The peak memory footprint matches single forward pass requirements
>     - Enables handling of 13B+ models within standard GPU memory constraints (weights can also be offloaded)
> 4. **Empirical Evidences**:
>     - For a 7b model trained on 8 * A800 with a batch_size=8 per GPU:
>         - GPU memory (initialization): **56,139MB**
>         - GPU memory (training):  **73,295MB**
>     - For a 32b model trained on 8 * A800 GPUs with a batch_size=1 per GPU:
>         - GPU (initialization): **64,821 MB**
>         - GPU (training): **66,381 MB**
>
> ## Q2. Motivation of Our Non-zero Initialisation Method
>
> Subtracting $A_0B_0$ from the pre-trained weights $W$ (as in PiSSA) is indeed feasible, this approach introduces critical practical limitations:
>
> 1. **Reproducibility Challenges in Inference:**
>     - Methods like PiSSA rely on randomized SVD to compute $A_0B_0$, making it impossible to exactly recover the same $A_0B_0$ during inference.
>     - Data-dependent methods (e.g., LoRA-GA) require training data to compute $A_0B_0$.
> 2. **Storage Overhead:**
>     - Storing either (a) the modified $W - A_0B_0$ or (b) $A_0B_0$ itself would resolve the above issue but at the cost of significant additional storage.
> 3. **Theoretical Limitation:**
>
> Considering there is a optimal downstream transfer $\Delta W^*$ for a downstream task to be tuned:
>
> $$\left\Vert AB - \Delta W^* \right\Vert_F^2 = \left\Vert W + (AB - A_0 B_0) - W^* \right\Vert_F^2 = \left\Vert AB - (A_0 B_0 + \Delta W^*) \right\Vert_F^2$$
>
> The minimal error of previous non-zero methods is given by $ A_0B_0 + \Delta W^* $ which is a biased best-r approximation of $\Delta W^*$. This bias can lead to sub-optimal solution.
>
> ## Q3. Optimal Solution for Decomposition
>
> We appreciate the reviewer’s question regarding our choice of optimization objective. To clarify:
>
> 1. **Optimality under L2:**
>     - The **truncated SVD** provides the optimal solution in L2 norm.
>     - we use the **pseudo-inverse** because:
>         - It naturally arises when solving the problem for optimal initialization given an initialized A. Solving this problem can achieve a compression result more aligned with the Eq. (4) in our paper.
> 2. **Why Not L1?**
>     - While L1 might offer sparser solutions, it lacks a closed-form solution and requires iterative optimization.
> 3. **Empirical Justification:**
>     - We tested a SVD-based initialization (similar to LoRA-GA):
>
>     | Method | GSM8K | HumanEval |
>     | --- | --- | --- |
>     | LoRA-GA | 71.39±0.90 | 43.29±0.71 |
>     | GoRA(SVD) | 72.68±1.06 | 44.71±0.93 |
>     | GoRA(pseudo-inverse) | 72.91±0.76 | 48.98±2.14 |
>
> Results demonstrate: Our method outperforms both SVD variants,  justifies our design choice.
>
> ## Q4. Dynamics change of rank
>
> Thank you for your insightful question. We appreciate this opportunity to clarify our methodology and share additional empirical findings.
>
> **Current Limitations in Dynamic Rank Adjustment**
>
> we fully acknowledge that the ideal rank distribution may evolve during training. However, implementing dynamic rank adjustment faces two fundamental challenges in modern training frameworks:
>
> 1. **Technical Constraints in Data-Parallel Training**: Frameworks like DeepSpeed flatten and distribute gradients/optimizer states across workers, making it *architecturally infeasible* to modify parameter shapes mid-training.
> 2. **Training Stability Concerns**: Even if rank adjustment were possible, it would require:
>     - Jagged learning rate scheduling
>     - Complex mechanisms to maintain optimizer state consistency
>     - Risk of destabilizing convergence
>
> Given these constraints, our current work focuses on ***pretraining-adaptive* rank allocation**
>
> **Empirical Analysis of Layer Importance Dynamics**
>
> We conducted a detailed study tracking layer importance every 5 steps under:
>
> | **Configuration** | **Details** |
> | --- | --- |
> | Model | Llama-3.1-8b-Base |
> | Dataset | MetaMathQA |
> | Optimizer | Adalomo [1,2] |
> | Batch Size | 8/GPU × 8 GPUs = 64 |
> | Total Steps | 1,024 |
> | Learning Rate | 1e-4 |
>
> **Key Findings:**
>
> 1. **Consistent Hierarchy**:
>     - **`wv`** layers maintain *significantly higher importance* throughout training
>     - **`wq`**/**`wk`** layers remain *persistently less important*
> 2. **Temporal Stability**:
>     - **Lower Layers**: Slight importance decay after ~200 steps
>     - **Middle Layers**: Moderate importance growth after ~200 steps
>     - **Upper Layers**: Remarkably stable importance trajectories
>     - **Exceptions**: Minority of layers show gradual monotonic changes (sum normalized importance ±<0.02)
>
> This suggests that while *minor* importance shifts occur, the *relative rank priorities* remain **largely stable** during the fine-tuning process.
>
> **References**
>
> [1] Adalomo (*ACL 2024*)
>
> [2] Full Parameter Fine-tuning for Large Language Models with Limited Resources. (*ACL 2024*)
>
> ## Q5. Rerun the initialization strategy
> We fully agree that periodically re-initializing the low-rank adapters to incorporate full fine-tuning training dynamics could further enhance performance. Below, we clarify the technical feasibility.
>
> ### **1. Technical Problem**
>
> **re-compressing gradients into low-rank adapters is possible**, though it requires tight integration with gradient collection pipelines (e.g., DeepSpeed). We are actively engineering this solution.
>
> ### **2. GoRA-pro: Integrating Fine-Tuning Dynamics**
>
> Motivated by the reviewer’s suggestion, we propose **GoRA-pro**, which refines initialization by simulating fine-tuning dynamics *before* formal training:
>
> 1. **Pre-Training Optimization**:
>     - Use the **AdaLomo** to perform *n*-step **updates** on the *full* model parameters.
>     - Store gradients at each step.
> 2. **Reinitialization**:
>     - Reset the model to its *original pre-trained weights*.
>     - Initialize low-rank adapters using the accumulated gradients.
>
> **Advantage over GoRA**:
>
> - **GoRA-pro actively integrates fine-tuning dynamics** into adapter initialization.
> - no prior or concurrent LoRA work explores such a hybrid of full fine-tuning and low-rank initialization.
>
> ### **3. Implications and Future Work**
>
> | Method | GSM8K | HumanEval |
> | --- | --- | --- |
> | GoRA | 72.91±0.76 | 48.98±2.14 |
> | GoRA-pro | 72.30±0.77 | 47.36±1.54 |
>
> Although currently GoRA-pro exhibits on-par performance compared to GoRA, we believe this direction could open new avenues for improving LoRA’s performance.
>
> ## Q6. How far from the best compression
>
> While it is challenging to directly compare the trained adapter matrices with the pre-training gradients, our initialization strategy  ensures that GoRA begins training with a low initial loss, implying effective capture of gradient information.
>
> To provide a rigorous and intuitive comparison, we conducted experiments on the **Llama-3-8B-Base** model trained on the MetaMathQA and Code-Feedback, evaluating the reconstruction error of initialized adapters against the full gradients across all adapted layers:
>
> - Math:
>     - Absolute Error: 0.0382
>     - Relative Error: 0.1147
> - code:
>     - Absolute Error:  0.0322
>     - Relative Error: 0.1152
>
> This indicates that our compression retains ~90% of the gradient information. Notably, when combined with the *n*-step accumulation strategy (see Q5), the initial loss further drops sharply from 0.68 to 0.44, underscoring the efficacy of our compression.
>
> ## Q7. MTBench Score
>
> We sincerely appreciate the reviewer’s observation regarding performance variations on MTBench. We would like to address this concern through the following clarifications:
>
> 1. **Nature of `r_ref` Parameter**:
>     - **`r_ref`** is fundamentally a *configuration parameter* rather than a tunable hyperparameter. Its primary role is to establish parity in total trainable parameters between GoRA and comparable methods (e.g., **`r_ref=32`** in GoRA corresponds to **`r=32`** in LoRA).
>     - We explicitly recommend practitioners use the largest computationally feasible **`r_ref`** value within their resource constraints.
> 2. **Understanding MT-Bench Variability**:
>
>     The observed performance fluctuations are consistent with well-documented phenomena in PEFT:
>
>     - The non-monotonic relationship between rank size and model performance on MT-Bench has been previously established (e.g., in LoRA-Pro and LoRA-GA).
>     - MT-Bench’s particular characteristics may contribute to this variability:
>         - Limited scale
>         - Possible distributional mismatch between training and evaluation data
>         - Known stochasticity inherent in LLM-as-judge evaluation
>
> 3. **Hyperparameter Management**:
>
>     While GoRA introduces four core parameters, we have additionally developed automated tuning mechanisms to enhance practical usability:
>
>     - Our response to Reviewer 4NWP’s Q4 details these optimization procedures
>
> ## Q8. Combine with QLoRA
>
> We are pleased to confirm that GoRA can indeed be seamlessly integrated with QLoRA's quantization approach.
>
> **Implementation Details:**
>
> - We quantize the pre-trained weights to **NF4 precision** for storage
> - During computation, we de-quantize these weights back to **bf16** precision
>
> **Empirical Results:**
>
> | **Method** | **GSM8K** | **HumanEval** |
> | --- | --- | --- |
> | QLoRA | 65.1 ± 0.95 | 43.49 ± 0.70 |
> | QGoRA | **70.58 ± 1.33** | **44.1 ± 3.68** |
>
> **Key Observations:**
>
> 1. QGoRA consistently outperforms QLoRA
> 2. The improvement is particularly notable on GSM8K
>
> ## Q9. Typo
>
> Thank you for remind us this typo! We have fixed this typo in the revised version of our paper.

---

> > ### Comment · Reviewer_zRT2 · 2025-08-05
> > **Thanks**
> >
> > I thank the authors for their rebuttal that addressed most of my concerns.
> >
> > I do, however, disagree on
> >
> > > Methods like PiSSA rely on randomized SVD to compute, making it impossible to exactly recover the same during inference.
> >
> > This is quite feasible by just storing a random seed. But this is minor, I've upgraded to accept :).

---

> ### Author Response · Authors · 2025-08-05
> **Thanks to Reviewer zRT2!**
>
> We sincerely thank the reviewer for their constructive feedback and for recognizing our efforts in addressing their concerns. We greatly appreciate their time and support in recommending acceptance.
>
> We are particularly grateful for the reviewer's insightful observation regarding storing the random seed for randomized method such as PiSSA! In response to this point, we would like to clarify that the official PiSSA repository (search PiSSA in github) currently provides two approaches to ensure consistency, as mentioned in our original submission: (1) storing the modified pre-trained weights, or (2) retaining the initialized low-rank weights. This observation lead to our conclusion in our rebuttal.
>
> We will add more discussions on these solutions (including storing the random seed) in our revised manuscript.
>
> Once again, we are grateful for the reviewer’s valuable input, which has helped improve our work.

---

### Official Review · Reviewer_4NWP · 2025-07-01

**Clarity:** 3
**Significance:** 3
**Originality:** 2
**Rating:** 4
**Confidence:** 4

**Summary:**

The paper proposes **GoRA**, a new method for setting Initialization and Rank in LoRA finetuning. GoRA allocates LoRA ranks before training by scoring each weight matrix with a gradient-based importance metric, and derives a data-driven, non-zero initialization for the B matrix. Empirically, GoRA seems to improve over LoRA, LoRA+, AdaLoRA, and several other baselines on language (GLUE, MT-Bench, GSM8K, HumanEval) and vision while keeping trainable parameters and GPU memory similar to that of LoRA.

**Questions:**

See above

**Ethical Concerns:**

["NO or VERY MINOR ethics concerns only"]

**Final Justification:**

During the author-reviewer discussion period, some of my concerns have been addressed. However, my main concern about compute cost of their method was not fully addressed. While I appreciate the authors willingness to implement more optimized approaches, I cannot factor that in the evaluation of the current paper since we do not have access to a revised version. I will therefore maintain my current score.

**Quality:**

3

**Strengths And Weaknesses:**

### I. Strengths

- **Unified treatment of rank and initialization** : GoRA re-uses the same gradient statistics to decide both how many LoRA ranks each layer receives and how the adapters are initialized. This avoids the masking overhead of AdaLoRA-style methods.

- **Robust gains across modalities** The reported result show that GoRA wins or competes with the strongest baseline on most tasks tested. With \(r_{\max}=128\) it even surpasses full fine-tuning on GSM8K and HumanEval, demonstrating that gradient-driven rank allocation scales to both language understanding and generation as well as vision classification.

- **Ablations**:   The paper provides ablations results on the effect of rank bounds, alternative importance metrics, etc. These ablations clarify why the chosen design works and how sensitive it is to hyper-parameters.


### II. Weaknesses

- **Compute start-up cost**: GoRA always requires N full forward-and-backward passes *before* training starts. While negligible for multi-hour jobs, this warm-up can dominate short experiments or hyper-parameter sweeps. Could N be selected adaptively—e.g., stop once the layer-wise importance estimates converge?

- **CPU off-load bottleneck**: Copying every gradient tensor to host memory each step may saturate PCIe on multi-GPU or FSDP setups. The paper shows single-GPU numbers only; bandwidth and wall-clock analyses for 2- and 8-GPU nodes would clarify scalability.

- **Host RAM usage**: Off-loading stores one fp32 copy of all targeted weights (about 20 GB for a 70 B model). Techniques such as bf16 accumulation or on-the-fly reduction could cap memory, but these are not discussed.

- **Hyperparameter sensitivity.**  GoRA introduces at least 4 other HPs $r_{min}, r_{max}, \gamma, N$. Performance can shift by ~3 accuracy points when the scaling factor $\gamma$ moves from 0 to 0.08. It is unclear how to set $\gamma\$ without an exhaustive grid search.

-  **Pseudo-inverse cost**: Unless I didn't see it, the authors do not discuss the compute cost of pseudo inverse. The initialization step requires computing $\bigl(A^{\mathsf T}A\bigr)^{-1}$ for *every* adapter. Although the matrix to invert is only $r \times r$, the cost scales cubically with rank ($\mathcal O(r^{3})$) *and linearly with the number of insertion points* (six per transformer block).  *For example*: on an 8 B model with 32 layers and six LoRA adapters per layer, moving from $r=32$ (≈ 32 k flops per inverse) to $r=128$ (≈ 2 M flops) increases total inverse cost by a factor of 64 -- non-trivial if performed on CPU.

### III. Suggestions for improvement

There are a couple of things that could help improve the paper.

1. **Adaptive or progressive probing.** Stop the warm-up when layer-wise importance estimates stabilize, or continue refining ranks asynchronously during early training.
2. **Guidelines for $\gamma$ and $N$.** Provide practical heuristics to avoid grid search.

---

> ### Author Rebuttal · Authors · 2025-07-29
>
> We sincerely appreciate the reviewers' recognition of GoRA's strengths, including its unified treatment of rank and initialization that avoids AdaLoRA-style masking overhead, its robust gains across multiple modalities (even surpassing full fine-tuning on GSM8K and HumanEval with rmax=128), and its comprehensive ablation studies. We also deeply value the reviewers' insightful concerns and valuable suggestions regarding computational efficiency (time/memory costs) and hyperparameter selection in GoRA. Below we provide detailed responses with experimental validations.
>
> ## Experimental Setup
> All experiments presented in this rebuttal were conducted on a computing node with:
> - **Hardware**:
>   - 8 × NVIDIA A800 80GB GPUs
>   - 105 × Intel(R) Xeon(R) Platinum 8336C @ 2.30GHz cores
>   - 1.875TB RAM
> - **Software Configuration**:
>   - model: Llama3.1-8b-Base
>   - Liger kernel for RMSNorm, fused linear cross-entropy loss, and MLP operations
>   - **Batch Configuration**:
>     - Global batch size: 64
>     - Per-GPU batch size: 8
>     - GoRA HP $N$: 8 (64 reported in our paper, where the per-GPU batch size is 1)
>     - Total number of samples for calculating gradients during GoRA initialization: 8 * 8 * 8 = 512
>     - No gradient checkpointing or accumulation
>   - Other configurations (including FlashAttention) match original paper settings
>
> ## Q1: Compute Start-up Cost
> We sincerely thank the reviewer for this insightful suggestion regarding initialization overhead. To address the computational cost of N full forward-backward passes, we have conducted additional exploration by implementing an adaptive convergence detection algorithm. This innovation automatically terminates gradient computation once layer-wise importance scores stabilize, significantly improving efficiency:
>
> | Method | Init Time (MetaMathQA) | GSM8K Accuracy | Convergence Criteria |
> |--------|-----------------------|----------------|-----------------------|
> | Fixed $N$=8 | 3min46.98s | 72.71±0.53 | - |
> | Dynamic (4 steps to converge) | 1min44.97s | 72.96±0.19 | <1% importance variation for 2 steps |
>
> **Impact**:
> - Full training time: 36min
> - Time saved: 7.5% total reduction
> - Accuracy maintained (±0.25 difference)
>
> ## Q2: CPU Off-load Bottleneck
> We thank the reviewer for this important observation. Our results indeed show the CPU offload overhead scales with GPU count:
>
> | GPU Count | Time/Step | Total Time |
> |-----------|-----------|------------|
> | 1 A800 | 3-4s | 23.6s |
> | 8 A800 | 15-20s | 3m46.98s |
>
> **Solution**: Implemented optimizations (detailed in Q3) to address PCIe bottleneck.
>
> ## Q3: Host RAM Usage
> We greatly appreciate the reviewer's valuable suggestions concerning memory efficiency. In response, we have implemented additional architectural optimizations that specifically target these memory concerns, yielding significant improvements:
>
> **Optimizations Implemented**:
>
> 1. **Memory-Saving Techniques**:
>    - Layer-wise immediate reduction with inter-GPU all-reduce
>    - Single-copy storage (rank-0 only) in CPU memory
>    - Rank-0 importance computation with broadcast synchronization
>
> 2. **Performance Improvements**:
>    - CPU memory: 8× → 1× (16-bit targeted modules' size)
>    - PCIe bandwidth: Reduced via 8→1 offload + broadcast
>    - Initialization time improvements:
>      - 8-GPU: 3m46.98s → 34.52s (4-5s per step)
>      - 8-GPU + early stop: → 20.32s
>
> We thank the reviewer again for these important suggestions, we will add a section and re-write the algorithm in the revised version of our paper.
>
> ## Q4: Hyperparameter Sensitivity
>
> We greatly appreciate the reviewer's highlighting of HP sensitivity. To minimize manual tuning, we propose these automated strategies:
>
> 1. **Rank bounds (r_min/r_max):**
>     - Auto-setting: **`r_max = 4×r_ref`**, **`r_min = 0.5×r_ref`**
>     - Guarantees: Maintains comparable parameters to LoRA (Table 7)
>     - Verification: Best performance across settings we tested (Table 5)
> 2. **Initialization steps (N) (new!):**
>     - Dynamic convergence:
>
>         a) Monitors inter-layer importance variation
>
>         b) Early stops when <1% change for 2 steps
>
>     - Outcome: Avg. steps reduced from 8→3 (2.8× faster)
> 3. **Scaling factor (γ) (new!):**
>     - Auto-selection:
>
>         a) Tests γ candidates ∈[1e-1,1e-5] during initialization
>
>         b) Selects γ minimizing first-step training loss
>
>     - Overhead: about 30s
>     - best γ searched for CodeFeedback: 0.0549 (5e-2 reported in the paper)
>     - best γ searched for MetaMathQA: 0.0858 (8e-2 reported in the paper)
>
> **Performance Impact**:
>
> | Method | GSM8K | HumanEval |
> |--------|-------|-----------|
> | Original | 72.91±0.76 | 48.98±2.14 |
> | Adaptive $N$ | 72.96±0.19 | 46.85±2.11 |
> | Adaptive γ | 72.5±0.38 | 50±3.23 |
>
> **Key Takeaway**: Automated approaches maintain performance (<±0.5 GSM8K, <±3 HumanEval) while reducing tuning effort.
>
> ## Q5: Pseudo-inverse Cost
> **Clarification**: All pseudo-inverse computations are GPU-based with minimal overhead:
>
> Below are initialization times for an 8B model across different ranks, measured using the following GPU-optimized implementation:
> ```python
>     def grad_compress_init(self,
>                       lr: float,
>                       scaling_by_lr: bool = False,):
>         if not hasattr(self.weight, 'grad_stored'):
>             return
>
>         # Convert weight_a to float32 on correct device
>         origin_weight_dtype = self.weight.dtype
>         weight = self.weight.to(torch.float32)
>         weight_dtype = weight.dtype
>         weight_device = weight.device
>         grad_stored = self.weight.grad_stored.to(weight_dtype).to(weight_device)
>         self.weight_a = nn.Parameter(torch.empty((self.lora_rank, self.in_features), dtype=weight_dtype, device=weight_device))
>
>         AT = self.weight_a.T
>         AAT = torch.matmul(self.weight_a, AT)
>
>         AAT_inv = torch.linalg.pinv(AAT + 1e-8 * torch.eye(self.lora_rank).to(weight_device)).to(weight_dtype)
>         AAT_inv_AT = torch.matmul(AT, AAT_inv)
>
>         # Compute weight_b using grad_stored (convert to float32 for computation)
>         weight_b_data = torch.matmul(grad_stored, AAT_inv_AT)
>
>         if scaling_by_lr:
>             stable_gemma = (lr / math.sqrt(self.lora_rank/self.in_features)) * (self.scale_rank)
>         weight_b_data *= (stable_gemma / self.scaling_alpha)
>
>         self.weight_b = nn.Parameter(weight_b_data.contiguous())
>
> ```
> | Rank | Time (8B model) |
> |------|-----------------|
> | 8 | 1.4s |
> | 32 | 1.56s |
> | 128 | 3.43s |
>
> Key Observations:
>
> - Even at rank 128 (requiring ≈2M FLOPs per inverse), the total initialization adds <4s overhead.
>
> - This represents <0.1% of typical training time, making the cost negligible in practice.
>
> We will clarify these implementation details and include benchmarking results in the paper. The GPU-based design ensures scalability, and we find no practical bottleneck from pseudo-inverse computations.
>
> ## Conclusion
> We sincerely thank the reviewers for these valuable suggestions. We will:
> 1. Add a detailed section on computational optimizations
> 2. Rewrite the algorithm description
> 3. Clarify pseudo-inverse costs
> 4. Include all experimental results in the revised paper

---

> > ### Comment · Reviewer_4NWP · 2025-08-06
> > **Thanks for the additional details.**
> >
> > Thanks for the additional details. Some of my concerns have been addressed, and while I appreciate the authors willingness to implement more optimized approaches, I cannot factor that in the evaluation of the current paper since we do not have access to a revised version. I will therefore maintain my current score.

---

> > > ### Author Response · Authors · 2025-08-06
> > > **Thanks to reviewer 4NWP again**
> > >
> > > We sincerely appreciate your time and constructive feedback. While we’re encouraged that some concerns were addressed, we fully respect your decision to maintain the current evaluation, as revisions cannot yet be implemented at this stage. Your insights on optimization are invaluable, and we look forward to incorporating them in the manuscript once the opportunity for revision arises. Thank you again for your thoughtful review.

---

> ### Author Response · Authors · 2025-08-05
> **Thanks to reviewer 4NWP!**
>
> We sincerely appreciate the time and effort Reviewer 4NWP has dedicated to evaluating our work. We are particularly grateful for the insightful suggestions regarding the communication/computation costs and HP sensitivity of GoRA.
>
> These constructive comments have guided us in further optimizing GoRA’s algorithm, significantly reducing initialization cost and HP tuning cost overhead—especially in multi-GPU scenarios. We hope our revisions have adequately addressed the concerns raised, and we would be happy to provide additional clarification or discuss any further questions the reviewer may have.
>
> Thank you once again for your valuable feedback.

---

### Decision · Program_Chairs · 2025-09-17

**Decision:**

Accept (poster)

**Comment:**

In this work, the authors propose a new PEFT method, called GoRA, which can be considered an upgrade to LoRA. While similar to LoRA, the method comes with two novel changes: a smarter initialization strategy of the low-rank matrices, and an adaptation of the rank during training based on the gradients. The method is able to outperform several PEFT methods such as rLoRA, LoRA+, AdaLoRA, PiSSA and DoRA (among others) in several standard language and vision benchmarks (GLUE, MT-Bench, GSM8K, HumanEval)  while keeping trainable parameters and GPU memory similar to that of LoRA.

The paper initially got all accept-regime scores, with 3 Borderline Accepts and 1 Accept. Some of the reviewers praised the method's robustness across modalities, ablations, the setting of experiments and evaluation, the performance, and computational efficiency preservation. However, they also requested some clarifications, mostly on regard to experimental section. In summary, the authors provided in the rebuttal the full experimental setup, compute start-up cost, CPU off-load bottleneck, RAM usage, hyperparameter sensitivity, the pseudo-inverse cost, gradient computing cost, optimal solution for decomposition, clarification with regards to dynamics change of rank, combination with quantization (GGoRA). In general, the rebuttal was extremely comprehensive, properly addressing most of the points raised by the reviewers. Saying that, the reviewer 4NWP says that their main concern about compute cost of their method was not fully addressed, but keeps their score, as do reviewers bxar and stk3. Reviewer zRT2 increases their score to Accept saying that the rebuttal addressed all of his concerns except a small one. Thus, the paper got final scores of 5, 5, 4, 4.

Having carefully read all the reviewers, I agree with the reviewers that this is a good paper above the acceptance threshold. I also agree that it is improving in an important and competitive baseline, and its very strong performance will benefit the community. Thus, I recommend the paper to be accepted, while urging the authors to improve their paper by integrating the changes from the rebuttal. Congratulations to the authors!